# A Shortcut-aware Video-QA Benchmark for Physical Understanding via Minimal Video Pairs

**Benno Krojer**[*]                                                            *benno.krojer@mila.quebec*
*FAIR at Meta*
*Mila*
*McGill University*

**Mojtaba Komeili**
*FAIR at Meta*

**Candace Ross**
*FAIR at Meta*

**Quentin Garrido**
*FAIR at Meta*

**Koustuv Sinha**
*FAIR at Meta*

**Nicolas Ballas**
*FAIR at Meta*

**Mahmoud Assran**
*FAIR at Meta*

[*]*Work done during internship.*

**Reviewed on OpenReview:** *https://openreview.net/forum?id=gvFgNJcSw1*

## Abstract

Existing benchmarks for assessing the spatio-temporal understanding and reasoning abilities of video language models are susceptible to score inflation due to the presence of shortcut solutions based on superficial visual or textual cues. This paper mitigates the challenges in accurately assessing model performance by introducing the Minimal Video Pairs (**MVP**) benchmark, a simple shortcut-aware video QA benchmark for assessing the physical understanding of video language models. The benchmark is comprised of 55K high-quality multiple-choice video QA examples focusing on physical world understanding. Examples are curated from nine video data sources, spanning first-person egocentric and exocentric videos, robotic interaction data, and cognitive science intuitive physics benchmarks. To mitigate shortcut solutions that rely on superficial visual or textual cues and biases, each sample in MVP has a minimal-change pair — a visually similar video accompanied by an identical question but an opposing answer. To answer a question correctly, a model must provide correct answers for both examples in the minimal-change pair; as such, models that solely rely on visual or textual biases would achieve below random performance. Human performance on MVP is 92.9%, while the best open-source state-of-the-art video-language model achieves around 40% compared to random performance at 25%.

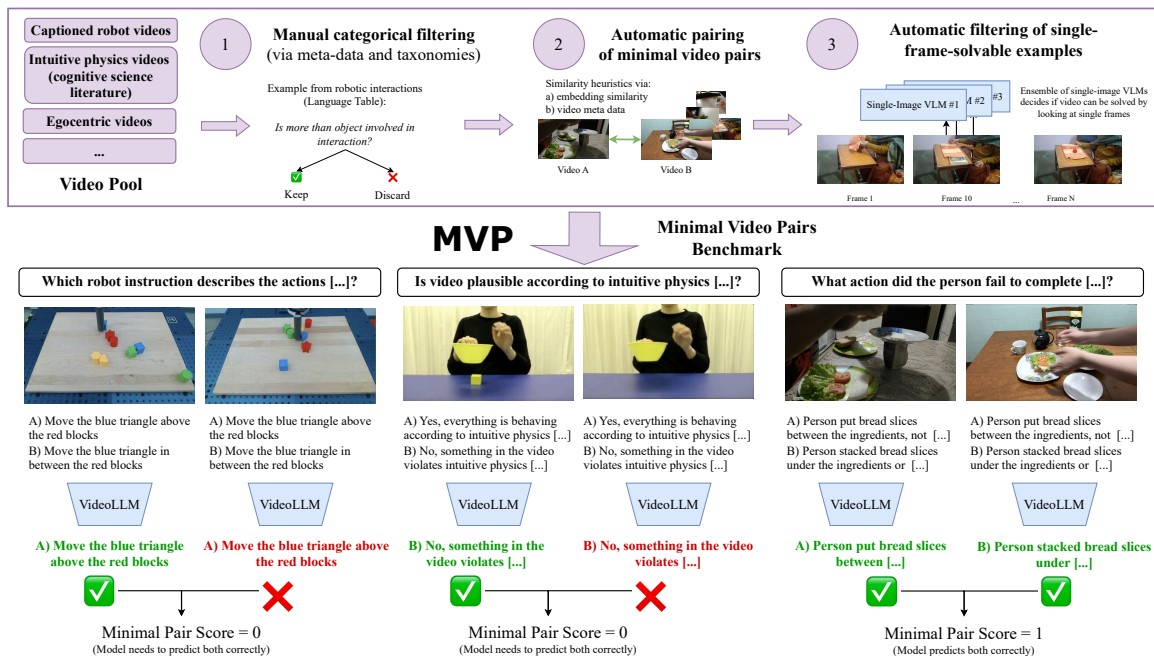

Figure 1: Illustrating MVP with its curation steps (top) and examples of our Minimal Pair Scoring (bottom).

# 1 Introduction

*Moravec's paradox* highlights a counterintuitive phenomenon: high-level reasoning tasks, often perceived as complex, are typically easier for AI agents to solve than sensorimotor and perception tasks, which are seemingly effortless for humans (Moravec, 1988).

Recently, large vision-language models have emerged as a promising paradigm for enabling perception capabilities in AI agents, demonstrating impressive progress on question-answering tasks across various domains including movies, documents, charts, and sciences (Alayrac et al., 2022; Team et al., 2024; Dubey et al., 2024; Wang et al., 2024a). This progress raises a natural question: do these models possess the spatiotemporal understanding and reasoning abilities essential for an agent to interact within the physical world, or do they buttress Moravec's paradox?

Various visual QA datasets have been proposed by the community to assess the spatiotemporal understanding of video-language models (Tapaswi et al., 2016; Maharaj et al., 2017; Li et al., 2024b; Patraucean et al., 2023; Zhang et al., 2023c; Xie et al., 2025; Wang et al., 2023c; Yi* et al., 2020); one of the most popular, MVBench (Li et al., 2024b), combines 11 video datasets into a single video QA benchmark.

While recent state-of-the-art video-language models obtain performance far superior to a random baseline on these benchmarks (Wang et al., 2024a; Shen et al., 2024; Li et al., 2024a), our investigation reveals that existing models can achieve strong performance on these tasks by relying on superficial visual or textual cues or biases. This is validated using simple baselines that discard the visual input or temporal aspect, yet achieve non-trivial performance. Similarly, recent work (Cores et al., 2024) shows that some of these tasks (Li et al., 2024b) fail to accurately measure the temporal understanding of a model.

In this work, we take inspiration from works in natural language processing (Levesque et al., 2012; Sakaguchi et al., 2021) and image processing (Thrush et al., 2022; Yuksekgonul et al., 2022) addressing visual and textual biases in evaluation, and introduce MVP, a video-QA benchmark containing minimal-change video pairs (MVP). Specifically, each video-question-answer sample in the benchmark is accompanied by a visually similar video possessing an identical question but an opposing answer (Figure 2). To answer a question correctly, a model must also provide the correct answer for its minimal-change pair while *processing them*

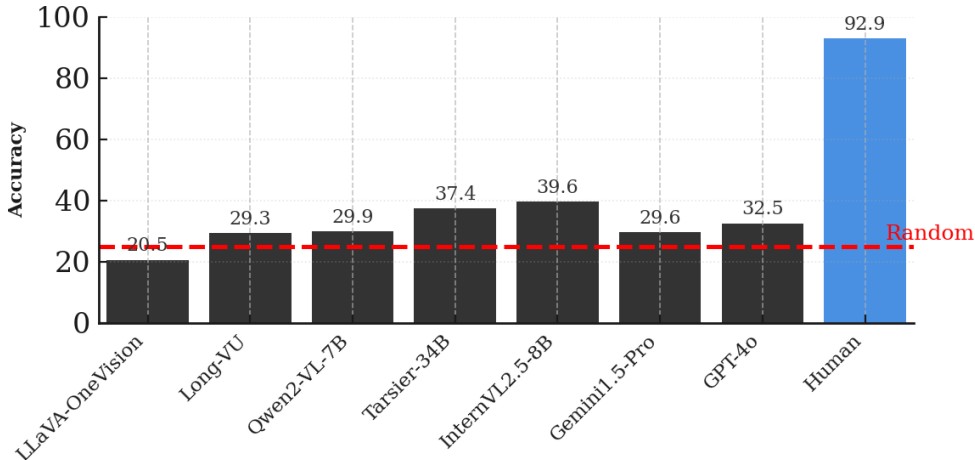

Figure 2: Performance of the strongest evaluated VideoLLMs on MVP (mini-version), compared to human performance.

*independently.* Many types of shortcut solutions are penalized under the minimal-pair scoring framework as a model relying on superficial visual or textual cues or biases would incorrectly output the same answer for both the samples in the pair.

While recent work created small sets of minimal-change video pairs for course-grained temporal reasoning (Zhang et al., 2024a; Liu et al., 2024), our key insight is that these pairs can be efficiently mined from existing video sources to test for several model capabilities through an automated process relying on visual embeddings and video meta-data. We propose an automatic process to find minimally different video pairs with limited human intervention, and then build these into a video-question-answer tuple with identical questions and opposing answers, enabling the scaling of the benchmark to a broad set of videos spanning diverse situations. We further process the mined samples using a model ensemble to filter out single-frame solvable examples — questions that can be answered using any single randomly sampled frame from the video — to encourage a stronger focus on video understanding. We build MVP by running our process on nine video sources spanning intuitive physics understanding, spatiotemporal reasoning, action anticipation, and robotic manipulation, leading to a total of $54,828$ multiple-choice video QA examples with minimal-change pairs; i.e. $27,414$ minimal-change pairs.

Next, we assess recent proprietary and open-source state-of-the-art video-language models using MVP. Specifically, we evaluate 2 closed-source models (GPT4-o (Achiam et al., 2023) and Gemini-1.5 Pro (Team et al., 2024)), and 7 open-source video-language models: LLaVA-OneVision (Li et al., 2024a), VideoChat2 (Li et al., 2024b), Mini-CPM (Yao et al., 2024), Qwen2-VL (Bai et al., 2023), Tarsier (Wang et al., 2024a), LongVu (Shen et al., 2024), InternVL-2.5 (Chen et al., 2024b). We find that even proprietary models are only slightly above random and that the best accuracy achieved across models is only at around $40\%$[1], in stark contrast to human baseline performance at 92.9% accuracy. These findings suggest that video-language models may still struggle with seemingly simple physical reasoning tasks, despite achieving relatively high accuracy on standard spatio-temporal reasoning benchmarks.

In short, we make the following contributions:

1. Analyze potential shortcut solutions on all 11 datasets in the popular MVBench (Li et al., 2024b) benchmark suite, using simple baselines consisting of language-only models, single-frame/image models, and Socratic LLMs.

2. Introduce MVP, a video QA benchmark for physical world understanding comprising minimally different videos — the largest of its kind by an order of magnitude with ~55K examples.

---

[1]40.2% on MVP-full and 39.6% on a smaller representative MVP-mini split we adopt to compare to expensive API models

3. Benchmark closed-source and open-source state-of-the-art models and identify a gap in physical world understanding; human performance on MVP is around 92.9%, while even GPT4-o and Gemini achieve around 30% compared to random performance at 25%.

We are publicly releasing MVP as well as a representative subset of the benchmark for faster inference (MVP-mini), together with a leaderboard and easy download scripts.

## 2 Robustness Analysis of Existing Video-QA Tasks

### 2.1 Shortcut solutions in video understanding

What is classified as a shortcut depends on the skills a task is meant to test, e.g., whether we deem certain features as *causal* or merely *spurious* (Geirhos et al., 2020); detecting and mitigating shortcut learning therefore remains an open problem (Geirhos et al., 2020). In our setting, we focus on shortcuts that models exploit in video tasks intended to test spatio-temporal or intuitive physics understanding. A standard way to reveal such shortcuts is to remove or alter parts of the input. For example, if a VideoQA model performs well with no video frames, then the task fails to measure for grounded visual understanding. Below, we investigate four such shortcut strategies, illustrated in Figure 3 with an example from the existing MVBench dataset (Li et al., 2024b) for each of the four strategies.

### 2.2 Empirical investigation

We begin by examining robustness of existing video QA benchmarks to shortcut solutions based on visual or textual cues or biases. Specifically, our analysis focuses on CLEVRER (Johnson et al., 2017), Perception Test (Patraucean et al., 2023), STAR (Wu et al., 2021), PAXION (Wang et al., 2023c), Moments in Time V1 (Monfort et al., 2020), FunQA (Xie et al., 2025), Charades-STA (Gao et al., 2017), MoVQA (Zhang et al., 2023b), NTU RGB+D (Liu et al., 2020), VLN-CE (Krantz et al., 2020) and TVQA (Lei et al., 2018), which are all included in the widely adopted MVBench (Li et al., 2024b) benchmark suite.

**Empirical Setup.** MVBench is comprised of 20 tasks from 11 datasets, collected in a multiple-choice video QA format, where a model is required to choose an answer $a_i$ from a tuple of question, video, and answer candidates $(q, v, [a_1, a_2, ..])$. Following standard practice (Goyal et al., 2017b), we study robustness to shortcuts by perturbing the task inputs, e.g., requiring the model to select an answer candidate without seeing the video or perhaps without reading the question, and compare to the accuracy achieved by a video LLM without perturbing the task inputs. We study 4 types of shortcut solutions by evaluating language-only models, video-only models, single-frame models, and simple Socratic LLMs. Results are reported in Table 1 using the original skill taxonomy outlined in MVBench.

**Language only.** Language-only models do not observe the video, and therefore select an answer candidate by only considering the textual inputs $q$ and the answer candidates $[a_1, a_2, ..]$. We leverage the Llama3-8B and Llama3-70B models due to their competitive performances (Dubey et al., 2024). In Table 1, we find that a Llama3-8B outperforms a random baseline by 6%, and a larger Llama3-70B outperforms a random baseline by 8%, suggesting that only a small subset of examples can be solved without considering the video input. However, digging into the individual datasets and sub-tasks in Table 1 reveals strong language-only performance on Action Antonym, where LLaMA3-70 achieves 78% compared to a random baseline at 50%. Upon closer inspection of the original dataset, we observe that many questions can be correctly selected by choosing the answer candidate with the highest marginal likelihood. For instance, given an example with answer candidates "book falling like a rock" versus "book rising like a rock," an LLM, just like a human, can rely on its language bias to infer that the former is probably the correct description without observing the video. We show a concrete example in Figure 3 where the LLM can rely on knowledge about TV shows without the video.

**Video only.** Video-only models do not observe the question, and therefore select an answer candidate by only considering the video input $v$ and the answer candidates $[a_1, a_2, ..]$. Table 1 shows that a video

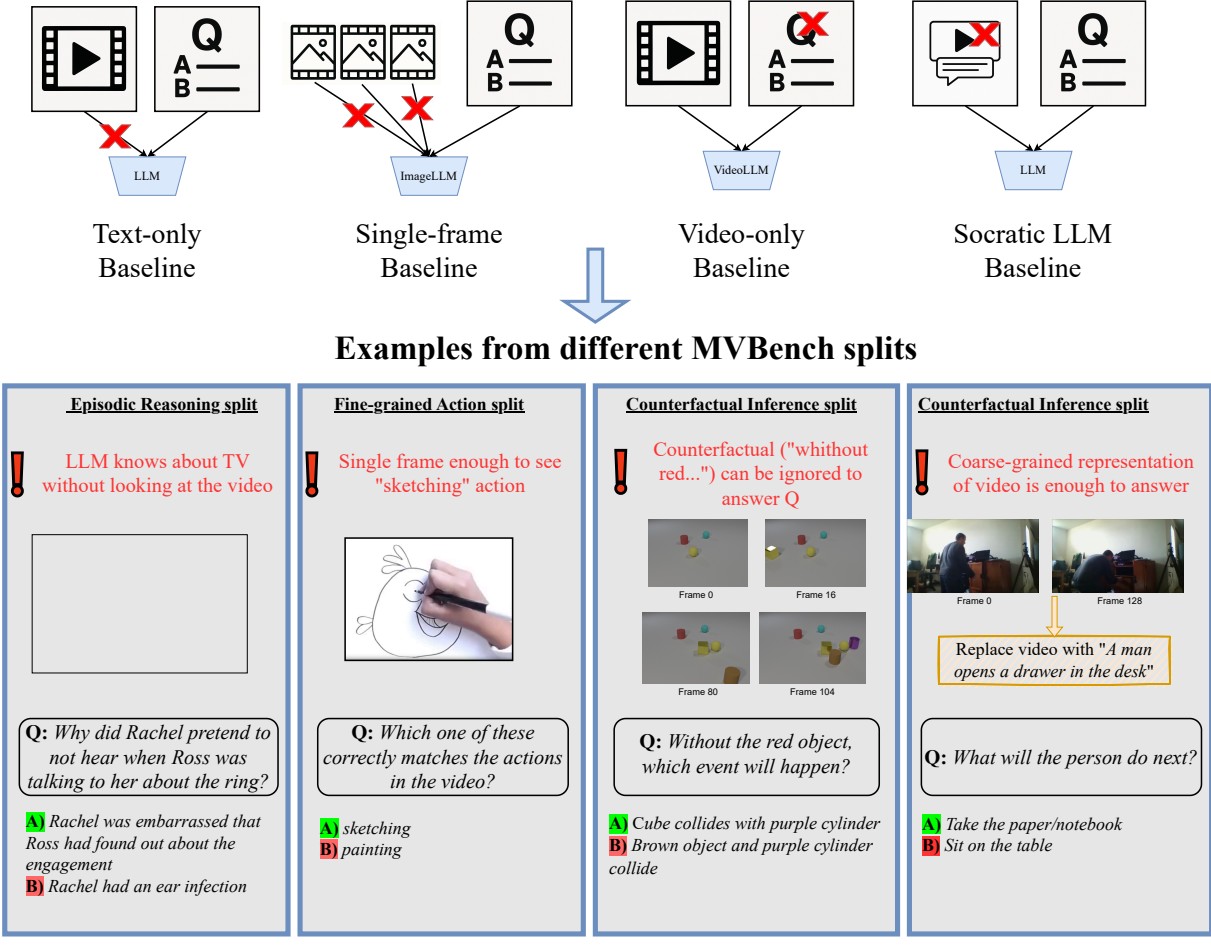

Figure 3: **An overview of four plausible shortcut baseline models might rely on for video QA tasks:** Ignoring the text (first), relying solely on single-frame understanding (second), ignoring the question (third), or relying on a task-independent coarse-grained representation of the image (fourth), i.e. a task-agnostic caption replacing the video itself.

LLM (VideoChat2-Mistral) can solve most sub-tasks without access to the question, reaching 50% overall accuracy; by comparison the same model achieves an accuracy of 61% when given the question in addition to the video, while a random baseline is at 30%. These findings indicate that the answer candidates for each question $[a_1, a_2, ..]$ are not sufficiently task-specific, as the model is able to discard the incorrect answers without knowing question. Recent work has found similar trends in language understanding QA benchmarks such MMLU-Pro (Wang et al., 2024c), where models are found to reach high scores without access to the question (Chandak et al., 2025).

This trend is particularly interesting on the counterfactual inference sub-task, where the counterfactual scenario such as "What happens if the red object is removed?" can only be known from the question. Manual inspection reveals that the correct answer in this task, e.g. "Cube collides with purple cylinder", (based on CLEVRER (Yi* et al., 2020)) often occurs in the video regardless of the counterfactual scenario, e.g., the two objects in question will collide regardless of the causal intervention. We visually illustrate this in Figure 3.

**Single-frame only.** Single-frame models do not observe the entire video, but rather are provided only a single frame $f_i \in v$ form the video. These models must therefore select an answer candidate by considering the frame $f_i$, the textual inputs $q$, and the answer candidates $[a_1, a_2, ..]$. We take $f_i$ to be the center

| Task | Avg | AA | AC | AL | AP | AS | CI | CO | EN | ER | FA | FP | MA | MC | MD | OE | OI | OS | ST | SC | UA |
|---|---|---|---|---|---|---|---|---|---|---|---|---|---|---|---|---|---|---|---|---|---|
| *Random Chance* | 0.30 | 0.50 | 0.33 | 0.25 | 0.25 | 0.25 | 0.31 | 0.33 | 0.25 | 0.20 | 0.25 | 0.25 | 0.33 | 0.25 | 0.25 | 0.50 | 0.25 | 0.33 | 0.25 | 0.33 | 0.25 |
| GPT-4V [†] | 0.44 | 0.72 | 0.39 | 0.41 | **0.64** | 0.56 | 0.11 | 0.52 | 0.31 | **0.59** | 0.47 | 0.48 | 0.23 | 0.12 | 0.12 | 0.19 | 0.59 | 0.30 | 0.84 | 0.45 | 0.74 |
| VideoChat2 (Mistral) | **0.61** | 0.86 | 0.37 | **0.44** | 0.55 | **0.76** | 0.72 | 0.49 | 0.36 | 0.40 | 0.50 | 0.64 | **0.88** | **0.69** | **0.49** | **0.87** | 0.75 | **0.41** | **0.85** | **0.50** | 0.62 |
| *Language only: Model considers question and answer choices, without access to the video.* | | | | | | | | | | | | | | | | | | | | | |
| Llama 3-8B | 0.36 | 0.63 | 0.38 | 0.27 | 0.25 | 0.28 | 0.35 | 0.43 | 0.29 | 0.43 | 0.29 | 0.29 | 0.38 | 0.27 | 0.21 | 0.46 | 0.29 | 0.36 | 0.52 | 0.40 | 0.52 |
| Llama 3-70B | 0.38 | 0.78 | 0.39 | 0.32 | 0.26 | 0.26 | 0.43 | 0.47 | 0.28 | 0.46 | 0.26 | 0.27 | 0.41 | 0.29 | 0.20 | 0.48 | 0.29 | 0.32 | 0.48 | 0.45 | 0.58 |
| *Video only: Model considers video and answer choices only, without access to the question.* | | | | | | | | | | | | | | | | | | | | | |
| VideoChat2 (Mistral) | 0.50 | **0.88** | **0.42** | 0.25 | 0.49 | 0.68 | **0.74** | 0.44 | 0.28 | 0.39 | **0.53** | **0.65** | 0.47 | 0.29 | 0.26 | 0.53 | **0.75** | 0.34 | 0.81 | 0.32 | 0.55 |
| *Single-Frame only: Model considers question, answer choices and a single key frame, without access to the full video.* | | | | | | | | | | | | | | | | | | | | | |
| Idefics3 | 0.47 | 0.72 | 0.37 | 0.31 | 0.52 | 0.48 | 0.42 | 0.54 | 0.31 | 0.48 | 0.40 | 0.44 | 0.55 | 0.42 | 0.34 | 0.49 | 0.50 | 0.37 | 0.73 | 0.48 | 0.60 |
| Qwen2-VL | 0.51 | 0.87 | 0.37 | 0.31 | 0.55 | 0.54 | 0.57 | **0.59** | **0.40** | 0.45 | 0.46 | 0.53 | 0.6 | 0.43 | 0.37 | 0.53 | 0.54 | 0.39 | 0.74 | 0.42 | **0.68** |
| *Simple Socratic LLM: Model considers the question, answer choices and a short generic description of the video.* | | | | | | | | | | | | | | | | | | | | | |
| Llama 3-8B | 0.44 | 0.56 | 0.38 | 0.28 | 0.49 | 0.57 | 0.35 | 0.53 | 0.29 | 0.42 | 0.30 | 0.35 | 0.56 | 0.42 | 0.32 | 0.50 | 0.56 | 0.35 | 0.68 | 0.44 | 0.56 |
| Llama 3-70B | 0.46 | 0.67 | 0.32 | 0.35 | 0.40 | 0.55 | 0.38 | 0.55 | 0.24 | 0.45 | 0.36 | 0.41 | 0.56 | 0.46 | 0.32 | 0.57 | 0.62 | 0.35 | 0.70 | 0.39 | 0.54 |

Table 1: **Shortcut Analysis on the 20 MVBench tasks from 11 datasets**: Optimal performance on these spatio-temporal reasoning benchmarks is frequently achieved by models relying on visual or textual biases (Single-Frame only, Video only, Simple Socratic LLM). [†]: GPT-4V accuracy from (Li et al., 2024b). Tasks: AA (Action Antonym), AC (Action Count), AL (Action Localization), AP (Action Prediction), AS (Action Sequence), CI (Counterfactual Inference), CO (Character Order), EN (Egocentric Navigation), ER (Episodic Reasoning), FA (Fine-grained Action), FP (Fine-grained Pose), MA (Moving Attribute), MC (Moving Count), MD (Moving Direction), OE (Object Existence), OI (Object Interaction), OS (Object Shuffle), ST (Scene Transition), SC (State Change), UA (Unexpected Action).

frame from the video and leverage Idefics3-8B (Laurençon et al., 2024) and Qwen2-VL-7B (Wang et al., 2024b) for the single-frame baselines. In Table 1, Idefics3-8B achieves an overall accuracy of 47% and Qwen2-VL-7B achieves an overall 51% accuracy, which is comparable to the performance of full-fledged VideoLLMs. Moreover, on Action Antonym, Action Prediction, Character Order, Egocentric Navigation, Episodic Reasoning, Fine-grained Action, State Transition, and Unexpected Action, the single-frame models are on par with (or even exceed) the performance of the VideoLLMs. Concurrent work (Cores et al., 2024) also studies the related bag-of-frame bias by shuffling the video frames.

**Simple Socratic LLM.** A Simple Socratic LLM (Zhang et al., 2023a; Zeng et al., 2023) replaces the video input $v$ with a short caption $c_v$ that can only convey a low-bandwidth description of the video. In practice, $c_v$ is 1 or 2 sentence-long caption generated by a separate VideoLLM (Zhang et al., 2024b) in a task-independent manner. The Socratic LLMs therefore select an answer candidate by only considering the low-bandwidth caption $c_v$, the question $q$, and the answer candidates $[a_1, a_2, ..]$. Following the text-only baselines, we use Llama3-8B and 70B. The performance of the Simple Socratic LLMs in Table 1 is significantly above random, with 44% for the LLaMA3-8B and 47% for the LLaMA-70B, suggesting that many sub-tasks (e.g. Character Order, Episodic Reasoning, Scene Transition) do not require fine-grained scene understanding.

**Summary.** The shortcut analysis reveals that existing models can often achieve strong performance on spatio-temporal reasoning benchmarks by relying on language cues (*Language only* shortcut) or visual cues, (*Video only* shortcut), and may not need to perform temporal reasoning (*Single-Frame only* shortcut), or possess fine-grained visual features (*Simplified Socratic LLM*). The first two baselines, text-only and single-frame, are what we one might call "hard requirements" for a video understanding benchmark: no example should be solvable in these settings. On the other hand, video-only (selecting most probably answer candidate without access to the question) and Simplified Socratic LLM (replacing video with a short caption) baselines are akin to "soft requirements": they might point to issues for certain examples for which you expect to require this information (e.g., MVBench's *counterfactual* sub-task should not be solvable without reading the question), but it does not imply a flaw in all cases.

## 3 Testing Physical World Understanding via Minimal Change Pairs

In this section we discuss the construction of MVP to mitigate shortcut solutions based on visual and textual biases. MVP is comprised of $54,828$ video QA examples covering various aspects of physical world

| | Domains of VideoQA-Examples | | | | | | | |
| Benchmark | Total | Natural Videos | Intuitive Physics | Robotics | Synthetic Videos | Minimially Diff. Videos | Procedural Single-Frame Bias Filtering | Format |
|---|---|---|---|---|---|---|---|---|
| CLEVRER | 76.3K | 0K | 21.4K | 0K | 76.3K | ✗ | ✗ | MC-QA |
| Perception Test | 11.5K | 11.5K | 0-0.2K | 0K | 0K | ✗ | ✗ | MC-QA |
| MVBench | 4K | 2.8K | 0K | 0.2K | 1.2K | ✗ | ✗ | MC-QA |
| TVBench | 2.5K | 1.9K | 0K | 0.2K | 0.6K | ✗ | ✗ | MC-QA |
| Vinoground | 1K | 1K | 0K | 0K | 0K | ✓ | ✗ | Group-Score |
| TempCompass | 0.5K | 0.5K | 0K | 0K | 0K | ✓ | ✗ | Group-Score |
| MVP | 54.8K | 22.3K | 9.9K | 25.8K | 32.6K | ✓ | ✓ | Pair MC-QA |

Table 2: **We compare with recent benchmarks that focus on similar skills**. Note that some videos may fall within several categories (e.g., synthetic intuitive physics videos). MVP contains minimally different videos at a much larger scale and across more diverse domains. From these benchmarks, MVP is the first to procedurally filter out examples due to single-frame bias. Group-Score = Present one video + two captions, and two videos + one caption. CLEVRER's intuitive physics entry is grayed as it only covers a narrow subset of intuitive physics concepts, largely based on collisions.

| Benchmark Category | Sources (# paired video-QA examples) | Example |
|---|---|---|
| Fine-grained human-object interactions | Perception Test (Patraucean et al., 2023) (3.5K), Something Something v2 (Goyal et al., 2017a) (3.6K) | **Q:** *What stops the motion of the object placed on the slanted plane after being released [...]?* **A)** Person or collision with another object **B)** High friction with surface |
| Fine-grained robot-object interactions | Language Table (Corey et al., 2022) (12.9K) | **Q:** *Which robot instruction best describes the actions in the video?* **A)** Move the green blocks in a vertical line below blue cube **B)** Move the green blocks and blue cube in a vertical line |
| Intuitive physics and collisions | IntPhys (Riochet et al., 2022) (0.2K), InfLevel (Weihs et al., 2022) (2.6K), GRASP (Jassim et al., 2024) (2.0K), CLEVRER (Yi* et al., 2020) (1.2K) | **Q:** *Is this video physically plausible/possible according to your understanding of e.g. object permanence, gravity, [...]* **A)** Yes, everything is behaving according to human intuitive physics understanding **B)** No, something in the video is off/strange or violates [...] |
| Coarse-grained temporal reasoning | STAR (Wu et al., 2021) (1.0K), Vinoground (Zhang et al., 2024a) (0.5K) | **Q:** *What is the best caption for this video?* **A)** The kayak flips over from facing upwards towards facing downwards **B)** The kayak flips over from facing downwards towards facing upwards |

Table 3: **Overview of MVP**. Each answer option A/B is correct for only one video in the minimal-change pair, while acting as a hard negative for the other video (by curation design). Note that we show the number of *paired* video-QA examples, thus the number of videos in our data is twice that amount.

understanding, including spatial reasoning, temporal understanding, human-object interaction, memory, counterfactuals, anticipation, and intuitive physics.

### 3.1 Task Definition

Designing MVP as a multiple-choice QA task ensures a simple unified setup that is easy to adopt by the community. At the same time, multiple-choice QA is highly flexible since most tasks can be reformulated as QA (several videos we source examples from were not originally designed for QA, such as Language Table). As a result any VideoLLM can be evaluated on MVP off-the-shelf. By contrast, other task formats, such as localization, have distinct advantages that may allow pinpointing the exact temporal or spatial failures (Cheng et al., 2025), however they do not allow the same ease of adoption and flexibility: applying a

VideoLLM to such task can require careful attention to the CoT prompts for different sub-tasks, and involve a mix of more complex metrics such as "mean visual Intersection over Union."

**Task formulation.** To improve robustness to the various shortcut solutions described in the previous section, we adopt a minimal-change pair approach (Levesque et al., 2012; Sakaguchi et al., 2021). An example in MVP consists of two video QA pairs $(q_1, v_1, [a_1, a_2])$ and $(q_2, v_2, [a_1, a_2])$ containing identical questions $q_1 = q_2$, visually similar videos $v_1 \sim v_2$, and two mutually exclusive (i.e., contradicting) answer candidates $a_1$ and $a_2$.

**Minimal-change Pair Scoring.** A model relying on superficial visual or textual cues or biases to solve a task will tend to produce the same output for each sample in the minimal-change pair. Thus, to penalize models for latching onto shortcuts, we only provide a positive score if the correct answer is produced for both minimal-change samples; the model receives each example $(q, v_1, [a_1, a_2])$ and $(q, v_2, [a_1, a_2])$ in isolation. Following a multiple choice QA framework, the model has to output a single answer letter (A or B) via task-specific prompts. In this setup, a random baseline achieves an accuracy of 25%.

### 3.2 Data Curation

**Question Taxonomy.** We wish to understand whether video LLMs possess the spatio-temporal understanding and reasoning abilities essential for an agent to interact within the physical world. As such we consider a coarse-grained taxonomy of question categories encompassing:

- Fine-grained human-object interactions,

- Fine-grained robot-object interactions,

- Intuitive Physics understanding,

- Coarse-grained temporal understanding.

We intentionally construct samples that are not overly reliant on cultural knowledge (Rawal et al., 2024; He et al., 2024; Li et al., 2024c) (e.g., movies) or specific domain knowledge (Tang et al., 2019) (e.g., detailed recipes) — tasks where language bias could contribute to the general performance.

We first manually filter videos from the sources described in Table 3 based on manual inspection (cf. Section C.1), then convert them into a question-answer format based on the associated meta-data (the textual captions for Language Table, the class labels for Something-Something-v2, QA annotations for PT, Vinoground, STAR, and CLEVERER, and the concept labels for IntPhys, InfLevel, and GRASP), yielding a starting set of 548K video QA examples.

**Minimal-change Pair Mining.** Next we procedurally identify minimal-change pairs from the 548K video QA examples produced from the previous stage. We note that 16% of the videos in our final benchmark ($\sim$ 8.8K examples) already possess explicit minimal visual pairs (even though most of these videos are not in a suitable format for video QA benchmarking, they can be converted into a minimal-change pair question-answer format based on the associated meta-data). For the remaining 84% of the videos, we leverage the following procedure to construct visual minimal-change pairs. In this process, we search for samples that have visually similar videos, identical questions (based on semantic matching), and contradictory answers. To then determine whether two videos with the same question are suitable minimal pairs, we use a) **symbolic and neural rules to determine video similarity** and b) **entailment detection** (Bowman et al., 2015; Dagan et al., 2013) between the correct answers of each video. Whether we rely more on symbolic or or neural rules of similarity depends on the data source: If a dataset has rich annotations (positions or attributes of objects) or structured captions (such as CLEVERER or Something Something-v2), we use hand-crafted rules and the NLP toolkit spacy (Honnibal & Montani, 2017) to narrow down the candidate pool of minimal pairs. This step would match videos with a large intersection of objects or attributes mentioned in the annotation/caption, leading to highly similar videos (e.g., the same objects appearing in both videos). Once

we have narrowed down the pool of candidate pairs, in the final step we rank video pairs by their cosine similarity in the ViCLIP (Wang et al., 2023b) video embedding space. We then select the top-ranked minimal video pairs such that each question or skill-type is sufficiently represented. At the same time, we ensure that the correct answers for samples in a minimal-change pair are sufficiently different, as the correct answer of one element in the pair must be a truly negative (*negative*) answer candidate for the other element, and vice versa: To avoid cases where both answers could be true at the same time (e.g., synonyms or more subtle cases) we define a set of textual rules to detect entailment for a subset of datasets. To illustrate this, in the *Fine-grained Robot-object interactions* category, our entailment-detection would discard the following pair of answers: A) "Move the blue cube towards the red heart" and B) "Move the blue cube to the left of the red heart", since A entails B. After this minimal-pair mining, we are down to 70K QA examples; cf. Section C.2 for technical details of the minimal pair mining process.

**Single-frame Bias Filtering.** Finally, to address single-frame bias, we remove examples that can be solved without the temporal information in the video; i.e., using only a single frame. We note that the input frame for this filtering stage should not be selected in a "smart way," since key-frame selection can be regarded as a basic form of temporal reasoning. In practice, five state-of-the-art multi-modal LLMs (LLama3.2-11B (Dubey et al., 2024), Molmo-7B (Deitke et al., 2024), Pixtral-12B (Agrawal et al., 2024), LLaVA-OneVision-7B (Li et al., 2024a), Idefics3-8B (Laurençon et al., 2024)) are prompted to answer the video-QA questions and "give their best commonsense guess given a single frame sampled from a video." If at least 4 out of 5 models in the ensemble predict the correct answer given the same frame, then we flag that frame as *solvable*. The minimal-change pair is then discarded if 30% of the frames in both videos are deemed solvable. This heuristic process removes around 20% of the samples from the previous stage.

**MVP Statistics.** We end up with $54,828$ examples in MVP, grouped into $27,414$ minimal-change video QA pairs. A breakdown of these examples is shown in Table 2 and Table 3 with a reasonably balanced split between natural videos, synthetic videos, robotics videos, and intuitive physics videos. An average video is 8.8 seconds long, the answer candidates contain an average of 8.1 words, and the datasets contains 2355 unique words in the questions and answers. Note that the word diversity is much less than MVBench (Li et al., 2024b), which has only 4K examples but twice the number of unique words (4338), reflecting our focus in testing for physical world understanding and not linguistically-diverse tasks with cultural or domain knowledge. Instead the task difficulty arises from the physical and perceptual aspects of MVP.

**MVP Data Quality Inspection.** In this section we briefly examine the data quality of MVP, i.e., how many examples are noisy or ambiguous. While we regularly inspected paired examples during pipeline design, it is important to also assess the final benchmark. Our overall human baseline of 92.9% (see Section 4) is a strong indicator: most paired QA examples are clean and solvable by humans.

On STAR, humans struggled for two reasons in rare cases: (i) the relevant action is sometimes not clearly visible (often in kitchen or living room scenes), or (ii) answer options in minimal pairs are too similar when the object is only partly visible. For example: Q: *What did the human pick up?* A) *The sandwich* B) *The dish.* Similarly, in Perception Test, rare cases were unsolvable due to ambiguous questions or because our answer matching produced two valid options. For instance, two candidate list might differ only in order but boil down to the same meaning: Q: *[...] What repeated actions did the person do?* A) *juggling, folding, [...]* vs. B) *folding, juggling, [...].* Finally, we also investigate on Language Table whether our hand-crafted rules for selecting minimal pairs works as intended: Out of 4 pairs of examples that human annotators had not successfully solved, one was simply human error while the other three were issues with the underlying source data itself. In the minimal pair of "*move the red circle left to green star*" and "*move the red circle diagonally top left to green star*", while the first caption mentions "left to green star" in reality the video showed more of "top left", thus making it ambiguous. We largely mitigate such non-exclusive candidates, as with Language Table's hand-crafted rules (see Appendix Section C). Still, some edge cases remain, as illustrated above.

## 4 Empirical Results on MVP

We evaluate several state-of-the-art open-source VideoLLMs on MVP, summarized in Table 4: LLaVa-OneVision (Li et al., 2024a), VideoChat2 (Li et al., 2024b), Mini-CPM-v 2.6 (Yao et al., 2024), Qwen2-

| Model | MVP (macro-avg) | Fine-grained human-object interactions | Fine-grained robot-object interactions | Intuitive physics and collisions | Coarse-grained temporal reasoning |
|---|---|---|---|---|---|
| Random | 25.0 (25.0) | 25.0 (25.0) | 25.0 (25.0) | 25.0 (25.0) | 25.0 (25.0) |
| *Any text model*[†] | 0.0 (0.0) | 0.0 (0.0) | 0.0 (0.0) | 0.0 (0.0) | 0.0 (0.0) |
| *Single-Frame Baseline (access to question, answer choices, and a single key frame from the video.)* | | | | | |
| LLaVA-OV (Qwen2-7B) | 11.8 (11.7) | 14.7 (12.2) | 8.7 (10.5) | 2.0 (2.3) | 21.6 (21.9) |
| Qwen2-VL (7B) | 16.7 (15.7) | 16.9 (13.6) | 20.1 (19.9) | 3.7 (4.4) | 26.3 (24.8) |
| InternVL2.5-8B | 19.1 (18.0) | 19.2 (15.1) | 18.3 (17.8) | 9.8 (11.7) | 29.3 (27.2) |
| *Video-only Baseline (access to video and answer choices, but removing the question.)* | | | | | |
| Qwen2-VL (7B) | 27.3 (26.2) | 30.8 (28.5) | 27.4 (25.8) | 6.0 (6.9) | 45.0 (43.5) |
| InternVL2.5-8B | 37.3 (35.1) | 38.9 (32.8) | 36.5 (32.9) | 16.0 (17.5) | 58.1 (57.2) |
| *Socratic LLM baseline (access to question, answer choices and a short generic description of the video)* | | | | | |
| Llama3-8b | – (19.7) | – (20.3) | – (27.3) | – (12.5) | – (18.9) |
| Llama3-70B | – (15.5) | – (16.9) | – (11.7) | – (10.6) | – (22.7) |
| *VideoLLMs (full access to the video, question, and answer choices.)* | | | | | |
| LLaVA-OV (Qwen2-7B) | 20.7 (20.5) | 24.3 (21.8) | 5.2 (5.2) | 5.8 (6.8) | 47.5 (48.2) |
| VideoChat2 (Mistral-7B) | 23.3 (22.0) | 25.7 (21.0) | 21.4 (20.1) | 10.1 (11.5) | 35.8 (35.3) |
| Mini-CPM-v 2.6 | 21.7 (20.2) | 21.3 (19.0) | 18.0 (14.1) | 9.2 (8.9) | 38.3 (38.7) |
| Qwen2-VL (7B) | 30.0 (29.9) | 27.1 (30.0) | 27.6 (25.7) | 20.0 (19.1) | 45.2 (44.7) |
| LongVU (LLaMA3-3B) | 20.6 (20.6) | 15.8 (14.1) | 14.8 (16.0) | 16.2 (16.7) | 35.4 (35.8) |
| LongVU (Qwen2-7B) | 29.9 (29.3) | 28.9 (26.3) | 21.5 (21.8) | 20.5 (22.3) | 48.6 (46.7) |
| Tarsier-7B | 26.0 (24.3) | 31.3 (24.5) | 18.7 (18.2) | 15.0 (16.3) | 38.9 (38.2) |
| Tarsier-34B | 38.8 (37.4) | **45.2** (38.7) | 36.3 (36.6) | 21.0 (22.1) | 52.7 (52.4) |
| InternVL2.5-8B | **40.2** (39.6) | 43.7 (37.8) | **40.2** (39.1) | **22.8** (23.9) | **54.4** (57.7) |
| Gemini-1.5 Pro | – (29.6) | – (43.1) | – (15.5) | – (19.6) | – (40.2) |
| GPT4-o | – (32.5) | – (36.1) | – (32.8) | – (16.2) | – (45.0) |
| Human | 92.9 | 91.3 | 91.7 | 97.6 | 90.9 |

Table 4: **Accuracy on MVP and MVP-mini in parentheses.** VideoLLM-performance is slightly greater than random chance, while humans achieve greater than 90% accuracy on all categories. Results for closed-source models are only shown on MVP-mini due to API costs. Performance is measured via Minimal Pair Score, wherein a model obtains a score iff the prediction for both QA examples of the pair is correct. [†]: If temperature of the LLM is zero, since from a text-side both examples in the minimal pair look the same.

VL (Wang et al., 2024b), Tarsier (Wang et al., 2024a) 7B/34B, LongVU (Shen et al., 2024), InternVL2.5-8B (Chen et al., 2024b), Gemini-1.5 Pro (Team et al., 2024), and GPT4-o (Achiam et al., 2023). Most notably these models differ in their generality: The models we evaluate are either generalist models (GPT4-o, Gemini 1.5), specialized for any visual inputs (LLaVa-OneVision, Mini-CPM, Qwen2-VL, InternVL), or specialized primarily for videos (VideoChat2, LongVU). We also consider the same baselines as in Section 2: text-only, single-frame, video-only (i.e. removing the question) and "Socratic LLM" (feeding a generic short/medium length caption to an LLM[2]). Note that we additionally evaluate on a smaller balanced version of MVP, dubbed MVP-mini, with around 1/3 of the original size.[3] For a fair comparison, we adopt the default parameters for all models, see Section H for details and for the full prompt to the models.

## 4.1 Main Results

**Overall performance of VideoLLMs.** Despite their strong performances on other video QA benchmarks (Li et al., 2024b; Liu et al., 2024; Mangalam et al., 2024; Xiao et al., 2021), Table 4 shows that most models perform around random chance (25% accuracy) with the exception of the Tarsier-34B model and InternVL2.5, reaching an average accuracy of 38.1% and 40.2 respectively. This is in contrast to human performances which obtain an average accuracy of 92.9% on a representative subset of MVP (cf. Section G).

While average performance is close to random for most models, we do observe non-trivial performance on several sub-tasks and data sources. In particular, VideoLLMs achieve better than random performance on

---

[2]We use captions from Qwen2-VL-7B after testing both InternVL2.5-8B and Qwen2-VL-7B as captioners
[3]We release MVP-mini for faster evaluation and lower costs of API models.

*Coarse-grained temporal reasoning*, meaning they possess some ability to distinguish the order of events in a video.

All models fall short on *Fine-grained robot-object interactions*, which involves understanding fine-grained object manipulation on a table with a robotic arm. This is particularly interesting given the proliferated usage of multi-modal LLMs for learning large-scale visuomotor control policies (Driess et al., 2023; Jiang et al., 2023). Most notably, the *Intuitive physics* category of MVP is by far the hardest with sub-random scores. As highlighted by previous works, intuitive physics concepts such as object permanence or gravity are easy for humans but remain hard for AI systems (Riochet et al., 2022; Jassim et al., 2024; Weihs et al., 2022; Du et al., 2023; Garrido et al., 2025; Bordes et al., 2025). While some methods have shown non trivial performance recently, LLMs in particular tend to achieve low performance (Garrido et al., 2025).

Finally we observe that shortcut-based baselines perform poorly on MVP (see upper rows in Table 4), confirming our efforts to mitigate shortcuts in the benchmark. Only the *video-only baseline* (Video LLM must pick the most likely answer candidate without access to the question) reaches close to the standard VideoLLM accuracy, e.g., InternVL2.5-8B reaches 39.6% on MVP-mini and 35.1% when removing the question (recall that random accuracy is 25%). However this only marginal performance degradation is mainly driven by the fact that accuracy on the *Coarse-grained temporal reasoning* split (sourced from STAR and Vinoground) remains stable in the video-only baseline: Here the question is indeed often not needed since both answer options are essentially captions. For many tasks the question might provide some context but is not strictly needed for capable models, e.g., "*What action is being performed in the video? A) running, B) walking*".

**Fine-grained failure analysis.** In this section we provide further insights where exactly models are falling short of human performance. We quantify failures on a more fine-grained level via subsets of MVP: Some sources in MVP are further divided into fine-grained splits by their original designers, where each split tests for a specific ability (e.g., object permanence, shape consistency, motion consistency, etc.). We show detailed model performance on the fine-grained splits in Appendix Section E, and discuss the most interesting cases here.

While performance on all intuitive sources is mostly below random, we find that some stronger models obtain non-trivial performance on four types of examples: *Gravity-Continuity* (LongVU (Qwen2-7B): 39.1%), *Unchangeableness* (LongVU (LLaMA-3.2): 35.2%, LongVU (Qwen2-7B): 42.2%); *Gravity-Support* (Tarsier-34B: 35.2%), *Continuity* (InternVL2.5-8B: 42.2%). We illustrate these numbers with two interesting examples: Figure 6 shows an example from *Unchangeableness* where both LongVU variants perform above random. However in another very simliar example in the *Unchangeableness* category shown in Figure 7, accuracy of all models is near or exactly 0%. Both examples showcase that objects cannot change color after being occluded for a short period of time. In the first example the ball is moving behind the wall while in second example the wall itself is moved up and down. These two examples highlight that even these rare success cases on intuitive physics are not particularly robust under slight perturbations to the setup.

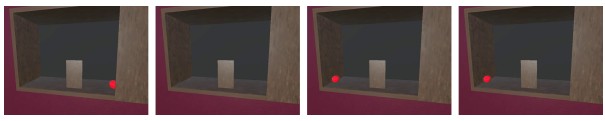
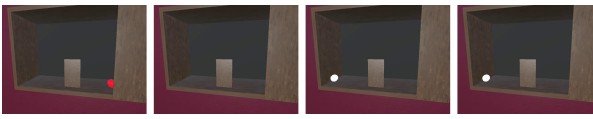

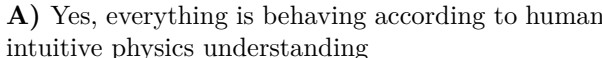

**A)** Yes, everything is behaving according to human intuitive physics understanding

**B)** No, something in the video is off/strange or violates human intuitive physics understanding

Figure 6: **One of the few intuitive physics categories where some models perform above random**: A MVP sample with videos sourced from the GRASP dataset. In the two videos the ball passes the central wall and either changes color (right video, violating intuitive physics) or remains the same color (left video).

Even some of the weaker models can achieve performance far above random on the MVP split *Fine-grained human-object interactions*, when looking closer into Perception Test categories such as *Counterfactual* (e.g., Qwen2-VL: 46.5%) and *Memory* (e.g., LongVU (Qwen2): 40.4%). We note that the questions in these two categories often involve significant symbolic skills next to the spatio-temporal aspect, such as reversing

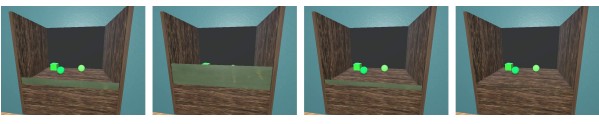

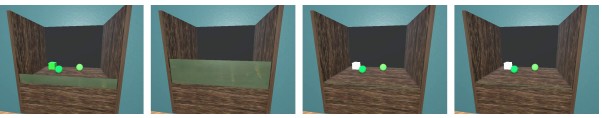

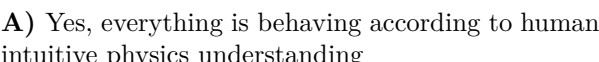

**A)** Yes, everything is behaving according to human intuitive physics understanding

**B)** No, something in the video is off/strange or violates human intuitive physics understanding

Figure 7: **One of the many intuitive physics categories where all models perform around 0%**: An MVP sample with videos sourced from the GRASP dataset. In the two videos a wall is moved up and down to either reveal that one object has changed color (right video, violating intuitive physics) or remains the same video (left video).

lists ("What if objects are put in reverse order?"), or OCR abilities. Thus if the spatio-temporal aspect is relatively straightforward, the LLM alone might get far with its abstract reasoning.

Finally, to the human eye examples where videos are sourced from Language Table and CLEVRER in MVP might look very similar (see appendix Figure 10 and Figure 12): They are both simple environments containing basic objects with canonical shapes. However, Language Table performance is generally around random while CLEVRER is above 50% in most cases; this is counterintuitive as CLEVRER questions should require more reasoning such as counting ("How many objects were moving at the end?") or prediction ("What will happen when the video ends?"). We offer several explanations: First, we observe that the video pairs in Language Table are closer to true minimal pairs; this is likely due to the fact that the Language Table video source is 10 times as large, making it easier for our pipeline to find hard minimal-change pairs. Second, due to the more complex reasoning in CLEVRER, there is also more room for biases to creep in. Third, Language Table videos carry subtle difficulties because it has "messier" distractor movements: humans teleoperate a robot arm and sometimes bump into other objects before taking the actual action. Finally, many VideoLLMs train on videos from the training split of CLEVRER and also the original CLEVR dataset (Johnson et al., 2017), while Language Table is a robotics dataset, and it is less likely that these robotics videos have crept into the training pipeline of common VideoLLMs.

### 4.2 Ablations and Analyses

**Importance of Data Curation.** In Table 5, we explore the effects of the minimal-change pair mining and single-frame bias filtering on model performance. For this exploration we use the smaller MVP-mini (see Section B) and report the average performance of five VideoLLMs [4].

When pairing videos randomly instead of using minimal-change pairs, the average accuracy across tasks is at 45.4%, far superior to random chance. Using minimal-change pairs, the average VideoLLM performance significantly drops to 27.3%. This result shows the importance of the minimal-pair framework and suggests that VideoLLMs can frequently leverage shortcut solutions or spurious features to solve QA tasks. Additionally, the average VideoLLM performance drops again by another 2.2% to 25.1% by removing single-frame solvable videos, with much larger drops on certain subsets. Note that while *Fine-grained robot-object interactions* and the *Intuitive physics and collisions* categories contain almost no single-frame biases, we can see significant drops of 3.5% and 3.3% for the other two categories (*Fine-grained human-object interactions* and *Coarse-grained temporal reasoning*) with this additional filtering step. Overall, Table 5 confirms that the minimal-change pair mining and single-frame filtering pipeline is effective at mitigating potential shortcut solutions in MVP.

**Influence of single-letter output format.** When evaluating VideoLLMs on MVP, we ask models to produce a single-letter answer in the format *Answer: A/B* (full prompt in Section H). This design keeps inference simple and uniform across diverse APIs and architectures. However one critique might be that models would benefit from open-ended output formats that encourage reasoning before answering. To test

---

[4]LLaVA-OV, VideoChat, Qwen2-VL, LongVU (Qwen2), Tarsier-7B

| Model | Overall | Fine-grained human-object interactions | Fine-grained robot-object interactions | Intuitive physics and collisions | Coarse-grained Temporal reasoning |
|---|---|---|---|---|---|
| | | *Pairing of random videos (with same question)* | | | |
| Avg. VideoLLM Acc. | 45.4 | 36.8 | 40.9 | 19.7 | 84.3 |
| | | *+ Pairing of minimally different videos* | | | |
| Avg. VideoLLM Acc. | 27.3 ↓18.1 | 28.7 ↓8.1 | 18.6 ↓22.3 | 16.7 ↓3.0 | 45.1 ↓39.2 |
| | | *+ Remove single-frame-solvable examples = final version of MVP* | | | |
| Avg. VideoLLM Acc. | 25.1 ↓2.2 | 25.2 ↓3.5 | 18.3 ↓0.3 | 15.2 ↓1.5 | 41.8 ↓3.3 |

Table 5: **We ablate the effect of our main curation steps.** Both the automatic pairing of minimal pairs and the single-frame-bias filtering lead to lower average model performance, with an especially large drop once we introduce the minimal pair setup.

this, we update the prompt for Gemini 1.5 to explicitly request reasoning before committing to a final answer (*Short reasoning* prompt):

> Short reasoning:
> Based on your observations, **reason about the following question [...] in 1–3 sentences**: {QUESTION + OPTIONS}. Then after the reasoning, select the best option [...]

Somewhat surprisingly, performance decreases, in line with recent work on output formatting in LLMs (Long et al., 2025), see Table 6.

To investigate further with a non-API model, we provide our strongest open-source VideoLLM (InternVL2.5-8B) with the same *Short reasoning* prompt variant and an additional step-by-step *Long reasoning* prompt:

> Long reasoning:
> Reason about the answer **step by step**, and **identify key positions of objects and the times of key events in the video**. Analyze the spatial relationships, temporal dynamics, and causal connections in the scene. Based on your detailed observations and step-by-step reasoning, address the following question: {QUESTION + OPTIONS}. After your detailed analysis, select the best option [...]

While performances does not decreases for InternVL2.5-8B, it remains the same (*Short reasoning*) or only increases minimally (*Long reasoning*), see Table 6. Overall, these results suggest that the primary bottleneck might not be symbolic reasoning but fundamental perception.

| Model / Prompt | MVP (macro-avg) | Human-Obj | Robot-Obj | Int. Phys. | Temp. Reason. |
|---|---|---|---|---|---|
| Gemini 1.5 (default) | 30.0 | 27.1 | 20.0 | 21.0 | 45.2 |
| Gemini 1.5 (+ Short reason) | 21.2 (-8.8) | 22.5 (-4.6) | 23.9 (+3.9) | 9.9 (-11.1) | 28.5 (-16.7) |
| InternVL2.5-8B (default) | 39.6 | 37.8 | 39.1 | 23.9 | 57.7 |
| InternVL2.5-8B (+ Short) | 39.6 (+0.0) | 37.6 (-0.2) | 38.8 (-0.3) | 24.2 (+0.3) | 58.0 (+0.3) |
| InternVL2.5-8B (+ Long) | 39.8 (+0.2) | 38.0 (+0.2) | 38.6 (-0.5) | 24.9 (+1.0) | 57.9 (+0.2) |

Table 6: **Effect of reasoning prompts on VideoLLM accuracy.** Default rows (gray) serve as baselines. Variants report raw accuracy and Δ relative to their baseline.

| Model | MVP (macro-avg) | Human-Obj | Robot-Obj | Int. Phys. | Temp. Reason. |
|---|---|---|---|---|---|
| Mini-CPM-v 2.6 | 57.5 | 57.3 | 52.5 | 52.7 | 67.5 |
| Qwen2-VL (7B) | 62.6 | 62.9 | 60.1 | 57.4 | 70.0 |
| InternVL2.5-8B | **67.7** | 67.1 | 66.1 | **59.8** | **78.0** |

Table 7: Single-example accuracy on MVP-mini.

| Model | MVP (macro-avg) | Human-Obj | Robot-Obj | Int. Phys. | Temp. Reason. |
|---|---|---|---|---|---|
| Mini-CPM-v 2.6 | 74.9 | 76.5 | 76.8 | 87.7 | 58.4 |
| Qwen2-VL (7B) | 65.6 | 65.7 | 68.7 | 76.5 | 51.4 |
| InternVL2.5-8B | 56.5 | 58.5 | 54.1 | 71.8 | 41.6 |

Table 8: Same-answer rate on MVP-mini.

**Additional metrics.** We further analyze MVP-mini with a representative set of open-source VideoLLMs, ranging from lower-performing (MiniCPM-v 2.6), to mid-level (Qwen2-VL 7B), to higher-performing (InternVL2.5-8B). Recall that our main metric, the *Minimal Pair Score*, assigns a point only if the model answers both examples in a pair correctly). We complement this with two auxiliary metrics: *single-example accuracy* and *same-answer rate* (the fraction of pairs where the model predicts the same option for both items).

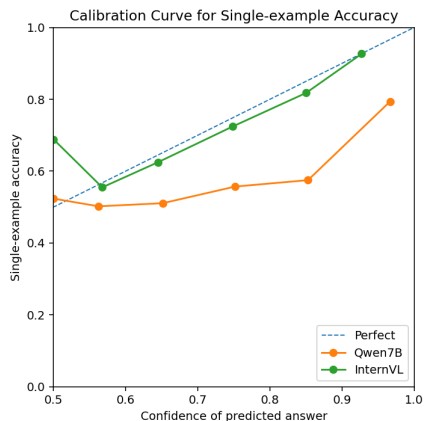

The latter captures a common failure mode: insensitivity to the minimal change, leading to identical responses for both videos. Both metrics are shown in Table 7 and Table 8. Single-example accuracy helps reveal that the Minimal Pair Score captures more than isolated correctness — it reflects whether models latch onto constant features across both videos. For instance, Qwen2-VL achieves a single-example accuracy of 62.6, an a Minimal Pair Score of 29.9, which is considerably lower, indicating that errors are not independent. Similarly, same-answer rate is far above "random chance" of 50% in most cases.

We also analyze the relationship between model confidence and single-example accuracy, plotting reliability diagrams for two representative models in Figure 8. For each question, we extract the probabilities of the answer tokens "A" and "B," interpret the probability of the predicted option as the model's confidence, and normalize e.g. $\hat{p} = \max\left\{\frac{p_A}{p_A+p_B}, \frac{p_B}{p_A+p_B}\right\}$. Note that this value is by definition at minimum 0.5, as shown in Figure 8. Binning these confidences reveals how well the self-reported probabilities align with empirical accuracy. InternVL2.5-8B exhibits particularly strong calibration, whereas Qwen2-VL tends to be over-confident in the mid-confidence range.

Figure 8: Calibration curve for two models (Qwen2-VL and InternVL2.5), plotting model confidence against actual accuracy.

**Relationship between accuracy and difficulty proxies.** We ask whether certain proxies of how difficult an example in MVP is negatively correlate with performance. For example on the vision side, we measure whether a longer video is harder for models on average. On the text side, we use two standard proxies, computed from the question and answer options (averaged): (i) Lexical frequency (Avg Zipf), using *wordfreq*'s[5] Zipf scale, where higher values indicate more common words; and (ii) Readability (Avg FleschKincaid grade[6]), where higher values indicate harder text (e.g. more words per sentence). We report Spearman correlations between single-example accuracy and each proxy (overall and per subtask). Overall, we find small or negligible correlations, except for the sentence difficulty, see Table 9. Moreover there is stark differences across the different data splits. Ultimately, there is no single easy proxy for task difficulty: We have seen that models do poorly on intuitive physics examples (Table 4), yet many intuitive physics examples are short videos. A concept like *object permanence* can be captured in a short video containing few objects, and does not need many words to be probed — yet models struggle.

| Difficulty Proxy | Overall | Human-Obj | Robot-Obj | Int. Phys. | Temp. Reason. |
|---|---|---|---|---|---|
| Video duration | -0.072 | -0.095 | -0.013 | -0.225 | 0.165 |
| Zipf | -0.024 | 0.039 | -0.033 | -0.102 | -0.134 |
| FleschKincaid | -0.204 | 0.055 | -0.002 | -0.244 | -0.139 |

Table 9: **Correlation of difficulty proxies with model accuracy. We compute correlations of proxies with model predictions aggregated from different models.**

---

[5]pypi.org/project/wordfreq/
[6]github.com/words/flesch-kincaid

## 5 Related Work

**Language biases in Vision-Language models.** Vision-and-language benchmarks, such as Visual Question Answering (VQA) (Antol et al., 2015; Goyal et al., 2017b; Marino et al., 2019) have been found to be vulnerable to language biases as evidenced by the performance of "blind" language-only models. Blind models are routinely shown to be efficient at solving many of the vision-and-language tasks (Goyal et al., 2017b; Zeng et al., 2023; Chen et al., 2024a), and can also solve several image-text retrieval benchmarks (Yuksekgonul et al., 2022; Hsieh et al., 2024) using language biases (Lin et al., 2023). Visual Question Answering in the video-language domain (Video-QA) (Li et al., 2024b; He et al., 2024; Xiao et al., 2021; Lei et al., 2018; Majumdar et al., 2024; Tapaswi et al., 2016; Rawal et al., 2024) also exhibits language biases, as shown in the performance of strong language-only baselines (Zhang et al., 2023a; Cores et al., 2024).

**Vision-centric biases in Vision-Language models.** State of the art vision-language models are shown to be surprisingly unaware of the vision inputs, where they often struggle with simple questions due to incorrect visual grounding (Tong et al., 2024), despite leveraging sufficiently powerful visual embeddings. VLMs are shown to be imprecise at spatial information understanding and geometry (Rahmanzadehgervi et al., 2024; Kamath et al., 2023). Similar biases exists in video-and-language tasks, where VideoLLMs typically exhibit single-frame bias (Buch et al., 2022; Lei et al., 2023) or spatial bias (Cores et al., 2024), where either a single frame is enough to solve the task, or the ordering of the frames is not important. To overcome this bias, benchmarks propose computing temporal certificate sets (Mangalam et al., 2024), key-frame bias (Buch et al., 2022), or investigate temporal understanding through shuffled frame inputs (Cores et al., 2024). In MVP, we operationalize a looser definition of temporal understanding for our filtering pipeline (Section 3) in that we keep an example if it is only solvable given the right *key-frame*, but discard it if it can be solved with any randomly sampled frame — the intuition being that key-frame identification can already involve temporal reasoning.

**Benchmarks addressing vision-and-language biases.** Several approaches are proposed in the literature to reduce the aforementioned biases in Vision-Language systems. One promising approach is to use minimally different pairs of inputs (Thrush et al., 2022; Yuksekgonul et al., 2022; Hsieh et al., 2024; Krojer et al., 2022; Wang et al., 2023a), also known as Contrast Sets (Gardner et al., 2020), which stem from related work in natural language processing (Levesque et al., 2012; Sakaguchi et al., 2021; McCoy et al., 2019). Minimally different input pairs restrict the models' abilities to use these biases, as *both* samples in the pair must be answered correctly to achieve a non-zero score. Similar to MVP, some highly adopted examples of such image-language benchmarks build on top of existing image sources (ARO (Yuksekgonul et al., 2022)), or fix them explicitly (SugarCREPE (Hsieh et al., 2024)). Commonly, the focus is on *textual* minimal-change pairs, e.g., providing several answer candidates for a question with only slight variations in word order (Yuksekgonul et al., 2022; Cores et al., 2024; Park et al., 2022; Li et al., 2023; Cai et al., 2024). However, even **textual minimal-change pairs can be susceptible to the same language biases** (Hsieh et al., 2024; Wu et al., 2023), which why we focus on visual minimal pairs in MVP. Other works, such as in Video-QA, focuses on visual minimal-change pairs. TempCompass (Liu et al., 2024) creates a small set of less than 0.5K artificial minimally different videos by manipulating the original video, e.g., playing the video in reverse, at a faster speed, or playing one video above the other. Vinoground (Zhang et al., 2024a) scrapes 0.5K minimally different video pairs from YouTube with the majority following the same pattern: *event A before B* vs. *event B before A*. Our work differs in several aspects from these (summarized Table 2), notably as well in terms of the scale of curation by showing that minimal video pairs can be procedurally extracted from existing video sources. While our Minimal Pair Score is inspired by Winoground (Thrush et al., 2022), unlike Vinoground, we intentionally do not adopt the Winoground metric directly since we want MVP to be agnostic to whether models can process several videos in one forward pass.

The language biases in existing vision-language benchmarks often stem from the over-reliance on world knowledge and plausible co-occurrences (Hsieh et al., 2024; Goyal et al., 2017b). Thus, MVP focuses on short videos with "basic" perceptual skills (spatial, temporal, or intuitive physics), which requires understanding of physical world properties (Yi* et al., 2020; Chen et al., 2022; Jassim et al., 2024; Riochet et al., 2022; Bear et al., 2021; Margoni et al., 2024; Baillargeon et al., 1985), reducing the space for blind LLMs to rely on their cultural knowledge.

## 6 Discussion and Limitations

Going back to our initial question, our results suggest that VideoLLMs do not yet perceive and understand the world as reliably as humans. After evaluating various state-of-the-art VideoLLM models for physical world understanding on MVP, the best model obtains only 40.2% average accuracy, while human performance is 92.9%. Yet, VideoLLMs are not completely blind. On some sub-categories of spatio-temporal understanding and intuitive physics, VideoLLMs can perform significantly better than random chance. Overall, our empirical evaluation shows that current VideoLLMs are still far from matching human performances on all tested tasks, calling for more research in this direction to develop better training data for world modelling, as well as novel learning criteria and model architectures. We anticipate MVP to help the development of the next generation of visual systems to perceive the world as robustly as humans.

**Limitations:** No benchmark comes without limitations. First, it is possible that more elaborate prompting strategies for free-form reasoning (CoT) and higher frame rates could improve performance. Additionally, using an automated curation approach will not be able to fully remove noisy examples; through manual inspection, we found some of the examples to be too simple, and a few others to be ambiguous, although we note that these noisy samples only represent a small subset of the overall data.

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

## Outline

**Section A** shows the license for each source in MVP and discusses legal aspects of the release.

**Section B** explains why and how we created a smaller version of MVP (MVP-mini).

**Section C** goes into all the nitty-gritty details of how we curated MVP.

**Section D** contains implementation details and further explorations for the shortcut analysis in Section 2 that motivated building MVP.

**Section E** focuses on the taxonomies from all the seed sources in MVP. We show more fine-grained results than our main results in Table 4 by showing performance categorized by these taxonomies (e.g. reasoning types in Perception Test).

**Section F** shows samples to provide readers with a realistic distribution of question and videos in MVP.

**Section G** explains how we arrived at the human baseline accuracy on MVP.

**Section H** provides details on how exactly we run inference, e.g. the exact prompt VideoLLMs (or single-image VLMs for the shortcut baselines) and how we extract answers.

**Section I (Behind the Scenes)** shows not just the final product (this paper) but also how we arrived here, what we discarded, and some personal reflections.

## A   Licensing and legal aspects

We only source videos for MVP with a permissible license. For legal reasons, we cannot directly release the videos that ended up in MVP after all curation steps. Instead we are providing an easy to run download script that goes over each source and downloads only the relevant videos. Similarly, when showing examples in the paper we ensure no faces of people are visible or other privacy concerns.

| Dataset | Source | License |
|---|---|---|
| Perception Test | `github.com/google-deepmind/perception_test` | Apache 2.0 |
| CLEVRER | `github.com/chuangg/CLEVRER/tree/master` | MIT / CC0 |
| STAR | `bobbywu.com/STAR/` | Apache 2.0 |
| InfLevel | `github.com/allenai/inflevel` | Apache 2.0 |
| IntPhys | `intphys.cognitive-ml.fr/` | CC BY 4.0 |
| Vinoground | `vinoground.github.io/` | Apache 2.0 |
| GRASP | `github.com/i-machine-think/grasp/tree/main` | MIT |
| CLEVRER | `clevrer.csail.mit.edu/` | CC0 |
| Language-Table | `github.com/google-research/language-table` | Apache-2.0 |
| Something-Something V2 | Qualcomm License Agreement PDF | Custom (Research Use) |

Table 10: Datasets with sources and licenses.

## B   MVP-mini

Next to the full MVP, we also release MVP-mini downsampled in a subset-aware manner to 18,290 video-QA examples (thus 9,145 pairs). MVP-mini will be faster to use, while MVP-full allows researcher to filter and curate derivatives at a large scale. For the most part our process simply involves selecting a random subset, except that we place additional conditions such that no dataset (or subset of a dataset) is underrepresented due to subsampling (i.e. never going below 500 examples). For example, IntPhys is already quite small with 360 videos so we don't remove any examples in this case.

# C   Details: The Curation of MVP

As illustrated in Section 3, we follow three main steps to curate MVP: 1) Manual categorical filtering 2) Automatic pairing of minimally different examples 3) Automatic filtering of single-frame-solvable examples. We also need to QAify 5 out of the 9 datasets.

While there are commonalities across datasets for how we implement Step 1 (categorical filtering) and Step 2 (pairing), there is also some differences that we describe here for reproducibility. Note that Step 3 is exactly the same across datasets, and is described in sufficient detail in Section 3. Hence, we focus on the first and second step.

## C.1   Manual categorical filtering

For 6 out of 9 datasets we select subsets and categories of questions suitable for MVP.

**Perception Test:** We manually annotate the 132 question types in Perception Test. Specifically we filter out question that either do not require temporal understanding ("Where is the person?") or are ill-defined. Around 20% of question types are discarded.

**Language Table:** We select the human-captioned and human-controlled split of Language Table which constitutes 440K. Thus we exclude other splits where the robot arm is automatically controlled and/or the robotic actions are synthetically captioned and not by a human. Additionally, we exclude any videos where the caption only mentions a single object such as "Move the arm to the left" to ensure complex enough interactions.

**CLEVRER:** We find an issue in the counterfactual split of CLEVRER and exclude these examples based on the meta-data associated with each video, e.g. object attributes and the exact position of each object at each frame. The issue is that most of the time, the object mentioned (*target object*) in the question ("What happens if the cube is not there?") is not actually involved in any collisions. As a result, the correct answer (e.g. "The red and yellow cube collide") is often depicted, whereas in a proper counterfactual example the correct answer should never be depicted but only happen in an "alternative world". Thus we filter out an example if the target object is never near any other moving objects (i.e. a collision) based on their coordinates. Moreoever, we filter out an example if the two objects mentioned in the correct answer are in fact already colliding in the video, based on their coordinates.

**InfLevel:** We only use the split with real-life videos where humans conduct the experiments in front of a camera, similar to experimental designs in psychology (Weihs et al., 2022).

**STAR:** We exclude the *Feasibility* and *Interaction* splits since they are often ill-defined, lead to strong language biases or are too easy.

## C.2   Automatic Pairing of Minimally Different Examples

We apply this step to only 5 out of the 9 datasets, since Vinoground (Zhang et al., 2024a) and the 3 intuitive physics datasets are already structured into minimal-change video pairs. Our pairing boils down to finding highly similar pairs, which we base on symbolic or visual similarity, and at the same time ensuring that both answers cannot be true at the same time, i.e. *mutually exclusive*. Especially the latter task, also known as *entailment* detection or *natural language inference*, has many nuanced edge cases.

This step is conducted on QA examples $x$ consisting of a question, a video, and answer candidates: $(q, v, [a_1, a_2, \ldots])$.

**Perception Test/STAR.**

1. Group QA examples into sets with the same question:

$$P = \{X \mid \forall x_i, x_j \in X, q_i \equiv q_j\}.$$

2. For a given $X$, examples $x_i$ and $x_j$ are grouped into potential pairs if they have opposite (*mutually exclusive*) correct answers:

$$P' = \{(x_i, x_j) \mid a_i \neq a_j\}.$$

3. From this set of potential pairs $P'$, we choose the top-$k$ for a given question based on visual similarity, measured via cosine similarity of embeddings from the video encoder ViCLIP-ViT-L (Wang et al., 2023b):

$$P_k = \big\{(x_{i_1}, x_{j_1}), \ldots, (x_{i_k}, x_{j_k}) \mid$$
$$\text{sim}(v_{i_m}, v_{j_m}) \geq \text{sim}(v_{i_{m+1}}, v_{j_{m+1}})\big\}.$$

In practice, we choose $k = 50$ for each question.

Additionally, we use dataset-specific rules after manual inspection, e.g., for the Perception Test, we require that for two potential pairs $x_i$ and $x_j$, neither correct answer $a_i$ nor $a_j$ is "Both the other options".

**Language Table.** Note that all examples in Language Table have the same question "Which robot instruction best describes the actions in the video?".

1. We group QA examples into sets such that a) both correct answers mention the same objects (e.g., both involve a "red triangle" and "green heart") and b) the set of tokens in $a_i$ and $a_j$ have a large enough overlap:

$$P = \big\{(a_i, a_j) \mid \text{obj}(a_i) \equiv \text{obj}(a_j) \wedge$$
$$0 < \text{token\_diff}(a_i, a_j) < 4\big\}.$$

Due to the finite number of attributes and objects in Language Table, $\text{obj}(\cdot)$ checks for these attribute and object key-words.

**Examples that would be matched:**

(a) move green star to the center left $\leftrightarrow$ move green star to the center Reason: i) all objects overlap, ii) set of words overlap except for one word ("left" only in first sentence)

(b) move green star to the top right of the red circle $\leftrightarrow$ move the red circle to the top of green star Reason: i) all objects overlap (just in different order), ii) set of words overlap except for one token ("right" only in first sentence)

2. We narrow this set of potential pairs $P$ with a visual similarity threshold, measured via cosine similarity of embeddings from the video encoder ViCLIP-ViT-L (Wang et al., 2023b):

$$P' = \{(x_i, x_j) \mid \text{sim}(v_i, v_j) > 0.9\}.$$

3. Finally, we ensure that answers are mutually exclusive, i.e., $a_i \not\Longrightarrow a_j$ and $a_j \not\Longrightarrow a_i$. In practice, this involves several hand-crafted rules after inspecting failure cases: If the order of objects mentioned is different, there is no entailment (e.g., "Move yellow triangle to blue heart" and "Move blue heart to yellow triangle"); if otherwise one answer contains a *general direction* such as "towards", "to" or "into" but the other answer contains a *specific direction* such as "left" or "above", there is entailment (we discard the example).

To illustrate, we would discard the following:

move the X towards Y $\Longrightarrow$ move X to the left of Y

move the X closer to the Y $\Longrightarrow$ move the X to the Y

move the X to left of Y $\Longrightarrow$ move the X to the top left of Y

**Something Something v2.** Note that Something Something v2 is a video caption dataset where each caption contains either one or two objects and a simple action, with in total 174 types of such actions. We QAify these examples with the question "Which action is being performed in the video¿' and use the caption with something-placeholders instead of objects as the answer $a$, e.g., "dropping something".

1. We group QA examples $a_i$ and $a_j$ into pairs such that the action in $a_i$ is a well-defined antonym of the action in $a_j$:
$$P = \{(a_i, a_j) \mid \text{antonym}(a_i, a_j)\}.$$
   In practice, we identify a subset of 82 action types (47% of all actions) that have a well-defined opposite, e.g., "spinning something so it continues spinning" and "spinning something that quickly stops spinning".

2. We narrow down pairs further by selecting a pair $x_i$ and $x_j$ if the videos contain the same object(s) based on their captions:
$$P' = \{(x_i, x_j) \mid \text{obj}(v_i) \equiv \text{obj}(v_j)\}.$$
   If no pairs fulfill this strict criterion, we relax it such that only one object must overlap:
$$P' = \{(x_i, x_j) \mid \text{obj}(v_i) \cap \text{obj}(v_j) \neq \emptyset\}.$$

3. From this set of potential pairs $P'$, we choose the top-$k$ based on visual similarity, measured via cosine similarity of embeddings from the video encoder ViCLIP-ViT-L (Wang et al., 2023b):
$$P_k = \big\{(x_{i_1}, x_{j_1}), \ldots, (x_{i_k}, x_{j_k}) \mid$$
$$\text{sim}(v_{i_m}, v_{j_m}) \geq \text{sim}(v_{i_{m+1}}, v_{j_{m+1}})\big\}.$$
   In practice, we choose $k = 4000$.

**CLEVRER.** Note that CLEVRER has detailed meta-data with a list of all objects throughout the video and their attributes (color, shape, material), with many videos featuring five or more objects.

1. Group QA examples into sets with the same question:
$$P = \{X \mid \forall x_i, x_j \in X, q_i \equiv q_j\}.$$

2. For a given $X$, examples $x_i$ and $x_j$ are grouped into potential pairs if they have opposite (*mutually exclusive*) correct answers:
$$P' = \{(x_i, x_j) \mid a_i \neq a_j\}.$$
   In the special case that the answers are both numerical, we require them both to be 1 apart, e.g., "How many objects are moving when the video ends? A) 2 B) 3".

3. We further filter the set $P'$ by requiring a large overlap of objects with the exact same attributes in both videos. Specifically, we keep a pair if the set of objects in $v_i$ is a "fuzzy subset" of the objects in $v_j$, or vice versa:
$$P'' = \{(x_i, x_j) \mid \text{fuzzy\_subset}(\text{obj}(v_i), \text{obj}(v_j)) \vee$$
$$\text{fuzzy\_subset}(\text{obj}(v_j), \text{obj}(v_i))\}.$$
   Here, $\text{fuzzy\_subset}(\cdot, \cdot)$ allows one mismatch between the sets of objects and their attributes. To illustrate with a simple example (most actual videos contain more objects):

   Video1 contains $\boxed{\text{Obj1 (red, triangle, metal), Obj2 (blue, triangle, metal)}}$

   and Video2 $\boxed{\text{Obj1 (red, triangle, metal), Obj2 (blue, triangle, rubber), Obj3(blue, cube metal)}}$.

   While Video1 is not an exact subset of Video2, it is almost a subset if we ignore that Obj2 does not match with any of the objects (which we call $\text{fuzzy\_subset}(\cdot, \cdot)$.

# D Details: Shortcut Analysis

We provide additional experiments for the shortcut analysis on MVBench datasets from Section 2.

**Language only shortcuts.** Similar to (Li et al., 2024b) we also tested a VideoLLM as a language-only baseline by blacking out the video (replacing it with zeros). With VideoChat2 this gave slightly worse results with both LLM-versions: 33.0% (Mistral) and 34.6 (Vicuna).

**Video only shortcuts.** For the video only we remove the question and only provide the answer candidates to the model. In detail, we tested several ways of removing the question (empty string, replace with "what?", explain to answer without question, etc) and found that replacing the question with "[REDACTED]" yielded the best performance.

**Single-frame only shortcuts.** First we study how performance varies when selecting frames at different positions: first, middle, last, random and finally key-frame. We choose a key-frame based on the highest CLIP similarity among all frames and all answer candidates (with the question prefixed to the candidate). We find that the performance differs only by 1-2% among these selection strategies except for the first frame which performed more than 5% worse than the rest with Idefics3. Since middle has the highest MVBench accuracy, excluding key-frame, we show middle frame results in the paper. In the main paper we show results for Idefics3 and Qwen2-VL, models that mostly focus on non-video tasks. We also test VideoChat2 variants but found performance to be worse, with either showing a single frame once or copying it 16-times as a "video".

**Simple Socratic LLM shortcuts.** In this shortcut we test how well models can still perform when the video is replaced by a much lower bandwidth representation and presented to a text-only LLM: a short generic caption of the video. We generate these captions with InternLM-XComposer-2.5-7B (Zhang et al., 2024b). We investigate how model performance differs when increasing the bandwidth of this caption from short, medium to long caption: "[...] Briefly describe this video in one sentence.", "[...] Describe this video in 1-2 sentences." and "[...] Describe this video in as much detail and length as possible.". We also ask whether focus on objects or actions helps, i.e. by prompting the captioning model to list the objects or actions in the video. While the long caption variant achieves the highest performance when provided to LLaMA3 8B and 70B, followed by action caption, we choose to show the medium caption variant (i.e. asking the model to caption the video in 1-2 sentences) in the main paper since this is most in the spirit of a short (1-2 sentences) and generic (not asking for anything specific) caption as a simple baseline.

**Additional robustness experiments beyond main paper.** As a sanity check, we also study how well a perplexity baseline and answer frequencies perform on MVBench. For the perplexity baseline we compute the perplexity of each answer candidate sequence based on LLM (LLaMA3-7B), i.e. how plausible this string is by itself. For example, are common objects or scenarios more often the correct answer? This would be reflected in such a baseline. However we find that this baseline performs around random overall. Next, we also compute statistics to determine if some subtasks of MVBench have a skewed distribution of answer frequencies, i.e. whether option A is more often correct than the other option B, C, etc. or if it is more often "yes" than "no", etc. Here we also also find very little evidence of any issues in terms of frequencies.

# E Details: Accuracies on sub-splits of each source in MVP

MVP is sourced from 9 seed datasets and 6 come of them with their own fine-grained taxonomy. In this section we provide a further breakdown of performance across these taxonomies.

For the other three sources (SSv2, Language Table, Vinoground), we provide performance in a single table:

TODO

| Model | Something Something v2 | Language Table | Vinoground |
|---|---|---|---|
| LLaVA-OV (Qwen2-7B) | 26.8 | 5.2 | 36.5 |
| VideoChat2 (Mistral-7B) | 39.1 | 21.4 | 16.9 |
| Mini-CPM-v 2.6 | 18.1 | 18.0 | 20.9 |
| Qwen2-VL (7B) | 32.6 | 27.6 | 16.5 |
| LongVU (LLaMA3.2) | 19.4 | 14.8 | 12.2 |
| LongVU (Qwen2-7B) | 38.1 | 21.5 | 25.6 |
| Tarsier-7B | 45.4 | 18.7 | 20.2 |
| Tarsier-34B | 65.4 | 36.3 | 34.2 |
| InternVL2$_5$ − 8$B$ | 52.8 | 40.2 | 39.5 |

Table 11: Accuracy on three sources of MVP that do not have further fine-grained splits.

**Perception Test.** The Perception Test dataset groups examples along two axes: *Skill Area* and *Type of Reasoning*.

**Skill Areas:** Memory (e.g. change detection seeing table early in video and at the end), Abstraction (e.g. counting), Physics (e.g. collisions), Semantics (e.g. distractor actions).

**Types of Reasoning:** Descriptive, Explanatory, Predictive, Counterfactual.

We show results of all models on these splits in Table 12.

**CLEVRER.** CLEVRER distinguishes into question types (as types of reasoning in Perception Test) and question subtypes, and we add *Moving Direction* as in (Li et al., 2024b):

**Types:** Descriptive, Explanatory, Predictive, Counterfactual.

**Subtypes:** Count, Exist, Query_Shape, Query_Material, Query_Color, Moving Direction.

We show results of all models on these splits in Table 13.

**IntPhys.** IntPhys contains three types of intuitive physics examples: *object permanence*, *shape constancy* and *spatio-temporal continuity*. We show results of all models on these splits in Table 14.

**InfLevel.** IntPhys contains three types of intuitive physics examples: *object permanence*, *shape constancy* and *spatio-temporal continuity*. We show results of all models on these splits in Table 15.

**GRASP.** GRASP contains much more categories than the other two intuitive physics datasets with a total of 15. We show results of all models on these 15 splits in Table 16.

**STAR.** As we adopted STAR, it has only two splits (since we removed two others in the filtering): Sequence and Prediction. We show results of all models on these splits in Table 17.

| Model | Memory | Abs-traction | Phy-sics | Sem-antics | Descrip-tive | Explana-tory | Predic-tive | Counter-factual |
|---|---|---|---|---|---|---|---|---|
| LLaVA-OV (Qwen2-7B) | 35.6 | 18.8 | 16.8 | 30.2 | 21.5 | 20.8 | 12.7 | 32.5 |
| VideoChat2 (Mistral-7B) | 25.1 | 7.4 | 12.2 | 15.4 | 9.8 | 18.6 | 14.5 | 22.8 |
| Mini-CPM-v 2.6 | 20.8 | 21.9 | 19.8 | 14.3 | 20.0 | 18.7 | 33.3 | 20.8 |
| Qwen2-VL (7B) | 30.9 | 18.7 | 18.1 | 29.4 | 21.2 | 17.4 | 19.1 | 46.5 |
| LongVU (LLaMA3.2) | 20.7 | 8.0 | 7.9 | 23.1 | 10.2 | 12.1 | 14.5 | 38.6 |
| LongVU (Qwen2-7B) | 40.4 | 15.0 | 13.9 | 29.7 | 18.1 | 20.6 | 13.6 | 43.9 |
| Tarsier-7B | 20.4 | 10.8 | 18.3 | 21.3 | 14.3 | 22.1 | 10.9 | 26.3 |
| Tarsier-34B | 32.0 | 18.5 | 24.9 | 32.4 | 21.5 | 32.7 | 24.5 | 41.2 |
| InternVL2$_5$ − 8$B$ | 29.2 | 29.5 | 29.0 | 27.2 | 30.9 | 27.7 | 33.3 | 29.2 |

Table 12: Accuracy for subsets of the Perception Test dataset.

| Model | Count | Exist | Query Shape | Query Material | Query Color | Moving Direction | Descriptive | Explanatory | Predictive |
|---|---|---|---|---|---|---|---|---|---|
| LLaVA-OV (Qwen2-7B) | 23.3 | 23.4 | 42.2 | 33.3 | 56.5 | 2.8 | 27.9 | 27.6 | 45.0 |
| VideoChat2 (Mistral-7B) | 50.4 | 71.9 | 65.6 | 69.8 | 68.5 | 10.2 | 57.2 | 13.8 | 80.0 |
| MiniCPM-v 2.6 | 17.1 | 34.4 | 37.8 | 12.7 | 60.2 | 4.6 | 28.7 | 17.2 | 60.0 |
| Qwen2-VL (7B) | 61.6 | 84.4 | 87.8 | 73.0 | 86.1 | 16.7 | 70.3 | 24.1 | 83.3 |
| LongVU (LLaMA3.2) | 45.1 | 14.1 | 34.4 | 19.0 | 26.9 | 13.0 | 38.6 | 6.9 | 26.7 |
| LongVU (Qwen2-7B) | 61.5 | 52.3 | 81.1 | 76.2 | 78.7 | 88.0 | 66.1 | 10.3 | 68.3 |
| Tarsier-7B | 56.0 | 70.3 | 77.8 | 65.1 | 81.5 | 14.8 | 62.4 | 31.0 | 83.3 |
| Tarsier-34B | 71.5 | 92.2 | 85.6 | 90.5 | 93.5 | 37.0 | 79.1 | 27.6 | 95.0 |
| InternVL2.5-8B | 81.5 | 93.8 | 88.9 | 96.8 | 96.3 | 99.1 | 88.1 | 24.1 | 93.3 |

Table 13: Accuracy for subsets of the CLEVRER dataset.

| Model | Shape constancy | Spatio-temporal continuity | Object permanence |
|---|---|---|---|
| LLaVA-OV (Qwen2-7B) | 0.0 | 0.0 | 0.0 |
| VideoChat2 (Mistral-7B) | 0.0 | 0.0 | 0.0 |
| MiniCPM-v 2.6 | 3.4 | 1.8 | 5.1 |
| Qwen2-VL (7B) | 14.7 | 4.7 | 6.8 |
| LongVU (LLaMA3.2) | 0.0 | 0.0 | 0.0 |
| LongVU (Qwen2-7B) | 14.7 | 2.3 | 4.5 |
| Tarsier-7B | 20.6 | 18.6 | 18.2 |
| Tarsier-34B | 5.9 | 0.0 | 6.8 |
| InternVL2.5-8B | 12.1 | 14.5 | 6.8 |

Table 14: Accuracy for subsets of the IntPhys dataset.

| Model | Continuity | Solidity | Gravity |
|---|---|---|---|
| LLaVA-OV (Qwen2-7B) | 0.0 | 0.0 | 0.0 |
| VideoChat2 (Mistral-7B) | 0.0 | 0.2 | 0.0 |
| MiniCPM-v 2.6 | 0.0 | 0.0 | 0.0 |
| Qwen2-VL (7B) | 11.3 | 8.8 | 13.3 |
| LongVU (LLaMA3.2) | 8.0 | 5.9 | 7.8 |
| LongVU (Qwen2-7B) | 7.0 | 6.8 | 9.6 |
| Tarsier-7B | 7.5 | 2.7 | 0.6 |
| Tarsier-34B | 5.0 | 14.0 | 14.5 |

Table 15: Accuracy for subsets of the InfLevel dataset.

| Model | Gravity Support | Continuity | Unchangeableness2 | Object Permanence | Inertia | Solidity continuity2 | Solidity continuity | Gravity Inertia | Unchangeableness | Inertia2 | Object Permanence2 | Gravity Inertia2 | Object Permanence3 | Collision | Gravity Continuity | Gravity |
|---|---|---|---|---|---|---|---|---|---|---|---|---|---|---|---|---|
| LLaVA-OV (Qwen2-7B) | 0.8 | 2.3 | 0.0 | 0.0 | 0.0 | 10.2 | 0.8 | 9.8 | 0.0 | 1.6 | 0.0 | 5.5 | 0.0 | 0.0 | 8.6 | 0.8 |
| VideoChat2 (Mistral-7B) | 0.0 | 0.0 | 0.0 | 0.0 | 0.0 | 0.0 | 0.0 | 0.0 | 0.0 | 0.0 | 0.0 | 0.0 | 0.0 | 0.0 | 0.0 | 0.0 |
| MiniCPM-v 2.6 | 3.1 | 10.9 | 10.2 | 0.8 | 0.0 | 0.8 | 4.7 | 5.5 | 6.2 | 6.3 | 0.8 | 3.9 | 3.1 | 2.3 | 23.4 | 14.8 |
| Qwen2-VL (7B) | 10.9 | 0.8 | 0.0 | 7.8 | 0.0 | 14.8 | 6.3 | 21.1 | 7.0 | 4.7 | 1.6 | 8.6 | 0.0 | 10.2 | 13.3 | 0.0 |
| LongVU (LLaMA3.2) | 0.8 | 0.8 | 35.2 | 0.0 | 4.7 | 0.0 | 0.8 | 0.0 | 2.3 | 0.0 | 0.0 | 0.0 | 3.1 | 0.0 | 24.2 | 2.3 |
| LongVU (Qwen2-7B) | 9.4 | 16.4 | 42.2 | 3.1 | 7.8 | 2.3 | 1.6 | 0.0 | 4.7 | 2.3 | 0.8 | 12.5 | 0.8 | 0.0 | 39.1 | 10.9 |
| Tarsier-7B | 10.9 | 10.2 | 5.5 | 0.0 | 3.1 | 8.6 | 3.9 | 7.3 | 3.1 | 3.1 | 1.6 | 0.8 | 0.0 | 0.0 | 3.1 | 11.7 |
| Tarsier-34B | 35.2 | 1.6 | 0.8 | 0.8 | 4.7 | 1.6 | 11.8 | 0.0 | 0.0 | 6.2 | 7.0 | 0.8 | 1.6 | 0.8 | 15.6 | 2.3 |
| InternVL2.5-8B | 10.2 | 42.2 | 3.1 | 22.0 | 19.5 | 9.4 | 2.3 | 13.3 | 6.2 | 9.4 | 4.7 | 6.2 | 3.9 | 3.1 | 3.9 | 3.1 |

Table 16: Accuracy for subsets of the GRASP dataset.

| Model | Prediction | Sequence |
|---|---|---|
| LLaVA-OV (Qwen2-7B) | 39.9 | 59.8 |
| VideoChat2 (Mistral-7B) | 33.3 | 53.3 |
| MiniCPM-v 2.6 | 54.3 | 54.0 |
| Qwen2-VL (7B) | 54.3 | 64.2 |
| LongVU (LLaMA3.2) | 52.2 | 50.5 |
| LongVU (Qwen2-7B) | 58.0 | 65.0 |
| Tarsier-7B | 39.9 | 50.7 |
| Tarsier-34B | 43.5 | 64.0 |
| InternVL2.5-8B | 67.4 | 75.5 |

Table 17: Accuracy for subsets of the STAR dataset.

## F    Illustrative examples from MVP

We show 2 examples for each of the nine sources in MVP, ranging from Figure 9 to Figure 17.

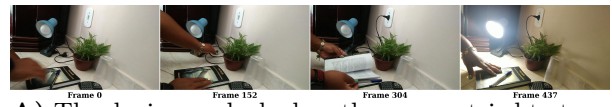

**A)** The device worked when the person tried to turn it on.

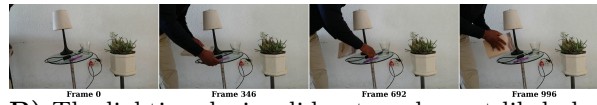

**B)** The lighting device did not work most likely because it was already unplugged at the beginning of the video.

**Question:** If the person tried to turn on the lighting device and it was not working, what was the reason for that?

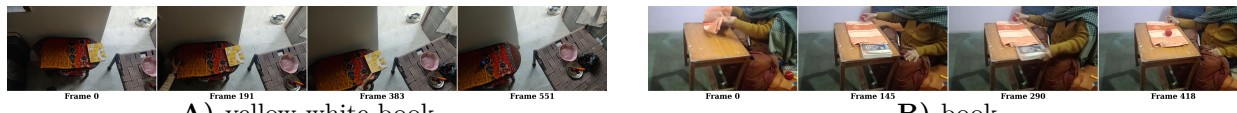

**A)** yellow-white book          **B)** book

**Question:** If the object were launched with a greater force or the friction was smaller, with which object would the launched object collide?

Figure 9: Two random samples (i.e. not cherry-picked) from the Perception Test split of MVP. We show pairs of videos where the answer for one video is the negative candidate for the other video.

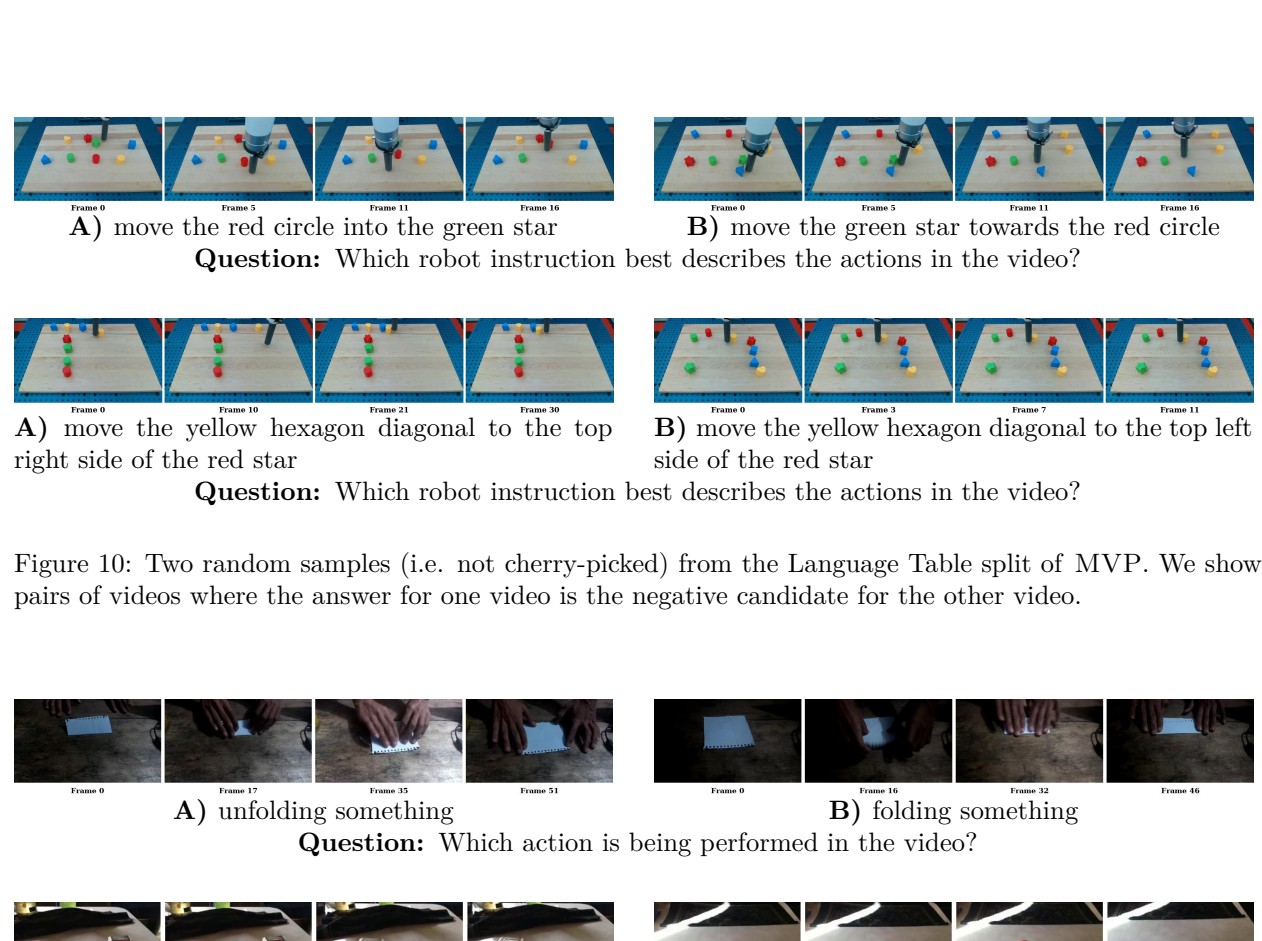

**A)** move the red circle into the green star     **B)** move the green star towards the red circle

**Question:** Which robot instruction best describes the actions in the video?

**A)** move the yellow hexagon diagonal to the top right side of the red star     **B)** move the yellow hexagon diagonal to the top left side of the red star

**Question:** Which robot instruction best describes the actions in the video?

Figure 10: Two random samples (i.e. not cherry-picked) from the Language Table split of MVP. We show pairs of videos where the answer for one video is the negative candidate for the other video.

**A)** unfolding something     **B)** folding something

**Question:** Which action is being performed in the video?

**A)** pushing something from right to left     **B)** pulling something from left to right

**Question:** Which action is being performed in the video?

Figure 11: Two random samples (i.e. not cherry-picked) from the Something Something v2 split of MVP. We show pairs of videos where the answer for one video is the negative candidate for the other video.

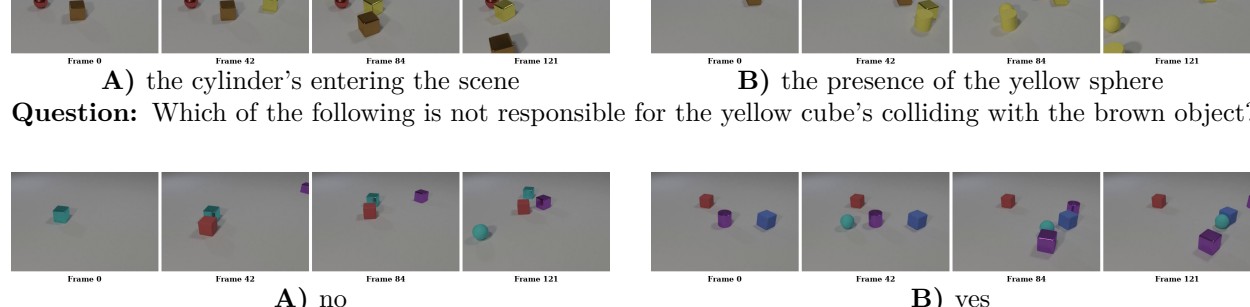

**A)** the cylinder's entering the scene     **B)** the presence of the yellow sphere

**Question:** Which of the following is not responsible for the yellow cube's colliding with the brown object?

**A)** no     **B)** yes

**Question:** Are there any stationary objects when the sphere enters the scene?

Figure 12: Two random samples (i.e. not cherry-picked) from the CLEVRER split of MVP. We show pairs of videos where the answer for one video is the negative candidate for the other video.

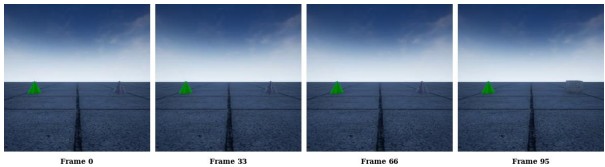

**A)** No, something in the video is off/strange or violates human intuitive physics understanding

**B)** Yes, everything is behaving according to human intuitive physics understanding

**Question:** Is this video physically plausible/possible according to your understanding of e.g. object permanence, shape constancy (objects maintain shape over time), continuous trajectories of objects? Assume it is the normal laws of physics. Your answer should be based on the events in the video and ignore the quality of the simulation engine. The rising wall is part of the experiment setup and should not be judged for plausibility.

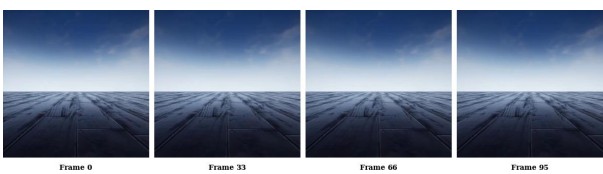
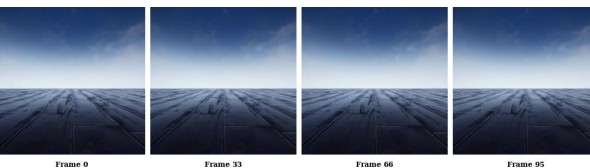

**A)** No, something in the video is off/strange or violates human intuitive physics understanding

**B)** Yes, everything is behaving according to human intuitive physics understanding

**Question:** Is this video physically plausible/possible according to your understanding of e.g. object permanence, shape constancy (objects maintain shape over time), continuous trajectories of objects? Assume it is the normal laws of physics. Your answer should be based on the events in the video and ignore the quality of the simulation engine.

Figure 13: Two random samples (i.e. not cherry-picked) from the IntPhys split of MVP. We show pairs of videos where the answer for one video is the negative candidate for the other video.

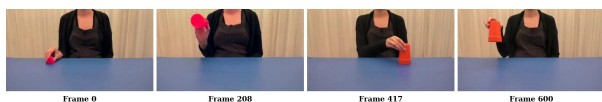

**A)** Yes, everything is behaving according to human intuitive physics understanding

**B)** No, something in the video is off/strange or violates human intuitive physics understanding

**Question:** Is this video physically plausible/possible according to your understanding of e.g. object permanence, gravity, trajectories of objects? (Assume it is the normal laws of physics and there are no magic tricks. Sometimes the video might start with visually showing the general setup of objects before the important actions happen)

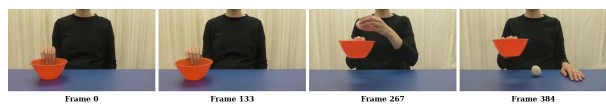
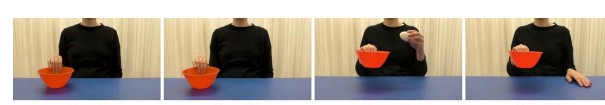

**A)** Yes, everything is behaving according to human intuitive physics understanding

**B)** No, something in the video is off/strange or violates human intuitive physics understanding

**Question:** Is this video physically plausible/possible according to your understanding of e.g. object permanence, gravity, trajectories of objects? (Assume it is the normal laws of physics and there are no magic tricks. Sometimes the video might start with visually showing the general setup of objects before the important actions happen)

Figure 14: Two random samples (i.e. not cherry-picked) from the InfLevel split of MVP. We show pairs of videos where the answer for one video is the negative candidate for the other video.

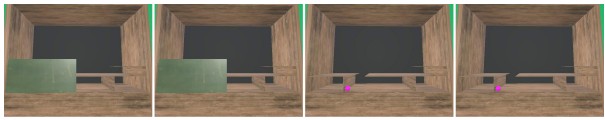

**A)** Yes, everything is behaving according to human intuitive physics understanding

**B)** No, something in the video is off/strange or violates human intuitive physics understanding

**Question:** The video you're seeing was generated by a simulator. Given how objects behave on earth, is the final position of the ball plausible? Your answer should be based on the events in the video and ignore the quality of the simulation.

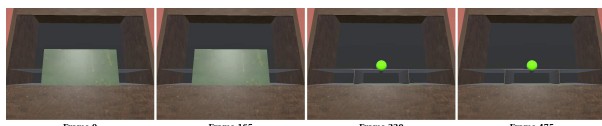

**A)** Yes, everything is behaving according to human intuitive physics understanding

**B)** No, something in the video is off/strange or violates human intuitive physics understanding

**Question:** The video you're seeing was generated by a simulator. Given how objects behave on earth, is the final position of the ball plausible? Your answer should be based on the events in the video and ignore the quality of the simulation.

Figure 15: Two random samples (i.e. not cherry-picked) from the GRASP split of MVP. We show pairs of videos where the answer for one video is the negative candidate for the other video.

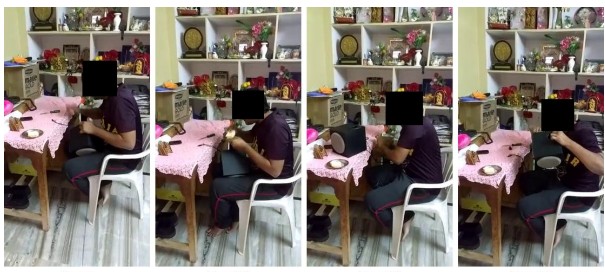

**A)** Took the sandwich.

**B)** Took the dish.

**Question:** What happened after the person sat at the table?

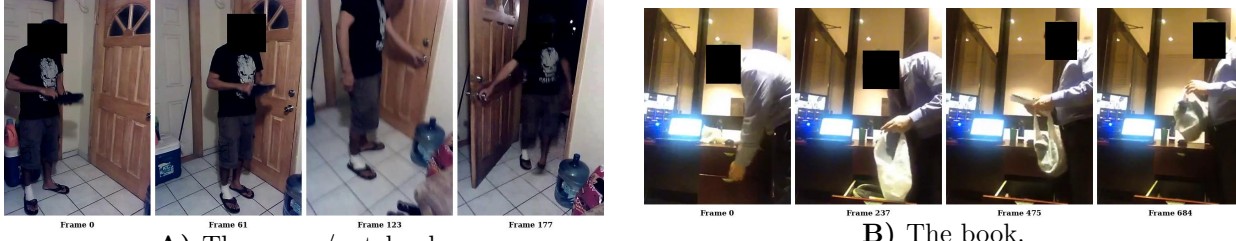

**A)** The paper/notebook.

**B)** The book.

**Question:** Which object did the person put down after they held the shoe?

Figure 16: Two random samples (i.e. not cherry-picked) from the STAR split of MVP. We show pairs of videos where the answer for one video is the negative candidate for the other video.

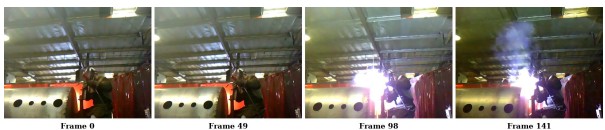 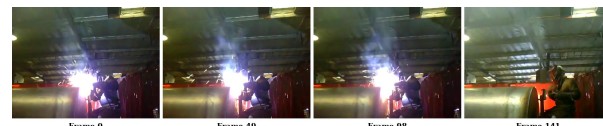

**A)** the man was not welding before he uses his welding gun and flames come out

**B)** the man uses his welding gun and flames come out before he was not welding

**Question:** What is the best caption for this video?

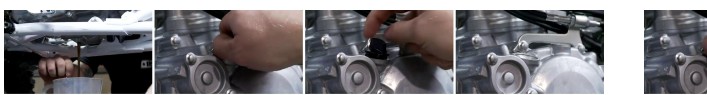 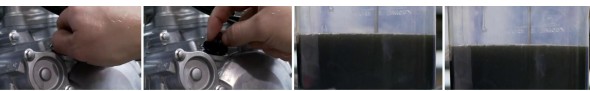

**A)** oil is drained before the cap is loosened

**B)** the cap is loosened before oil is drained

**Question:** What is the best caption for this video?

Figure 17: Two random samples (i.e. not cherry-picked) from the Vinoground split of MVP. We show pairs of videos where the answer for one video is the negative candidate for the other video.

## G  Human Annotation

We assigned over 600 videos to 6 researchers from our lab, recorded their answer responses, and then computed the benchmark metric using the pair-wise scoring. Each person was assigned one video from the pair at a time, thus avoiding any advantage over VideoLLMs that would come from seeing both videos in a minimal-change pair (i.e., avoiding any knowledge that the answer can only be AB or BA).

In addition to the 4 main splits of MVP, we also provide a breakdown of human accuracy across the 9 data sources we used:

- Perception Test: 87.8%

- Something Something V2: 94.7%

- Language Table: 91.7%

- CLEVRER: 100%

- InfLevel: 100%

- IntPhys: 96.3%

- GRASP: 94%

- STAR: 81.8%

- Vinoground: 100%

## H  Details on prompting multi-modal LLMs

For a fair comparison (HuggingFace, API), we use each model's default inference setup (e.g. greedy decoding for all models) which are quite similar anyways. For example, videos are similarly processed (resolution, number of frames) and then fed through CLIP or ViT-based video encoders. Most models use either 8 or 16 frames by default, e.g. Tarsier-34B uses 16 which we believe is enough for most MVP examples, yet the model is still far below humans.

We use the following prompt for all VideoLLMs (filled with an example question from our benchmark):

---

**VideoLLM Prompt**

You are an expert video understanding AI system. Carefully watch the video and pay attention to the cause and sequence of events, the details and movements of objects, and actions of people. Based on your observations, select the best option that accurately addresses the following question:
Q: {Question}
A) {Correct answer for video1}
B) {Correct answer for video2}
Even when unsure, always answer with a single letter from A or B, format exactly like: 'Answer: A/B'.

---

We extract the answer letter via a simple regex and find that this approach fails in only less than 1% of examples.

# I Behind the Scenes

In this section we go beyond what usually goes into a paper and discuss how the paper came about, what did not work, or what motivated the authors - so in essence: all the things that are usually deemed too subjective or "unscientific", yet would help other researchers, especially those joining the field, often much more than the polished narrative of the main paper.

## I.1 Motivation and timeline

Several of the authors who work on video modeling felt a growing frustration with existing benchmarks that often rewarded the wrong things. So the direction of the project was quickly set after a short period of brainstorming: Quantify in what ways existing benchmarks are broken, and then fix it. We then spent a few months staring at hundreds of examples from the MVBench datasets, scouting for glaring issues or shortcuts and manually annotating lots of data. First, we tested the simple baselines with respect to frequency, text-only or single-frame biases, and soon included the less often discussed video-only (remove question) and Simple Socratic LLM shortcuts. In between we had philosophical discussions about benchmark design (bottom-down vs. top-down) or what it means for a video task to be truly temporal: is it temporal if two-frames are needed, or if a single frame is needed but it has to be a key-frame (needle in a haystack), or ...? Regarding benchmark design, should we adopt other paper's taxonomies or design or own? Should one collect all kinds of examples and ad-hoc define a taxonomy (bottom-up), or should one define a taxonomy, then systematically collect examples to fit the taxonomy (top-down)? From the beginning the idea of minimal video pairs generated excitement among us: Minimal visual pairs have led to much progress in the field of vision-and-language compositionality (e.g. Winoground), yet had not been explored much in the realm of moving images.

There were, and perhaps still are, plans to crowd-source human shortcut performance, i.e. how much better are humans at solving video tasks when given single frames? At scale this could also be used to filter out examples with more precision than our five-model ensemble approach. Doing human crowd-sourcing well is not trivial, it is time-intensive and requires dedication but it can lead to much stronger insights than relying purely on automatic metrics and black-box models.

After the exploration phase, we executed on the benchmark building: From the start we had identified several promising datasets to mine minimal video pairs and continuously added a new source roughly every week. Perception Test was the first to go through our curation pipeline and hence took the longest as we were still refining the pipeline steps. Language Table was very hard to do well with many edge cases in the entailment detection, and also with its scale of 440K video-caption examples (imagine looking for potential pairs of videos, i.e. $440,000^2$ combinations).

## I.2 Observations and lessons learned

1. There are too many edge cases to catch every single one at this scale of data curation.

2. Paper writing is smooth when the story and contribution is clear from the beginning of the project (this was not the case in the first author's last paper so it was nice to observe the contrast).

3. The intuitive physics datasets are (to the subjective taste of the first author) the cleanest and most fascinating sources in MVP, even though they are not directly suitable for video QA, and only make up a small fraction of the benchmark.

4. At the same time, Something Something v2 truly stood the test of time as a great video understanding dataset due to its low noise-ratio at a scale of 200K examples, and coverage of interesting yet simple phenomena.

5. Frame rate plays a big role for solving many examples in MVP. In order to push the field further we are now asking the models more and more nuanced questions, and the answer may lie only in a short span of a less than second. However many models may not have access to this short span in principle as they represent a video as 16 uniformly sampled frames.

6. Parsing outputs from LLMs into a structured format such as answer options can feel like the wild west sometimes: Could models perform better if we prompt them better, or have more flexible ways of extracting the answer?

### I.3 Advice for others working on a similar direction

Video-QA is becoming an increasingly popular topic; it is a very exciting direction with enough dimensions for everyone to innovate on: long video benchmarking, intuitive physics, social common-sense/Theory-of-Mind/narratives, novel simulation engines, and so on.

Despite our best efforts studying shortcuts, we probably missed some shortcuts or issues in MVP. It is good to think two steps ahead what kind of shortcuts future more capable models could take. It is easier said than done, but in retrospect older video benchmarks from 2015-2020 might have at least been able to address single-frame biases, a priori, during benchmark design. Instead now the field took years to identify and clean up benchmarks.

