# OpenReview forum: "A Shortcut-aware Video-QA Benchmark for Physical Understanding via Minimal Video Pairs"
_TMLR — Accepted by TMLR_

### Review · Reviewer_4W8E · 2025-08-04

**Summary Of Contributions:**

## Summary
This paper first evaluates a few shortcuts in existing video QA benchmarks. It reveals that some of current benchmarks suffer from language shortcut, video shortcut, single-frame shortcut, and course-grained caption shortcut. Then, the paper introduces Minimal Video Pairs (MVP), a video QA benchmark that mitigates the shortcuts. Eash sample in MVP includes a pair of similar videos with the same question but opposite answers. The videos are sourced from nine existing video QA benchmarks under four categories. And the video pairs are matched by an automatic procedure. The paper evaluates eleven multimodal LLMs and the best performance is slightly better than random chance and significantly worse than human performance. Especially, on the intuitive physics and collisions category, all the models perform worse than random chance.

## Strengths:
1. The paper reveals four types of shortcuts in existing video QA benchmarks, providing insights about these benchmarks to the community.
2. The proposed MVP benchmark avoids the language shortcut by collecting video pairs and mitigates the single-frame shortcut by filtering.
3. Although MVP benchmark requires paired videos, they search for the paired videos from existing benchmarks in an automatic procedure. This makes it easier to scale up the MVP benchmark and results in 27k video pairs.
4. The paper conducts comprehensive evaluations on eleven multimodal LLMs, including open-source and proprietary models. Human performance is also provided as a reference.

## Weaknesses:
1. While the proposed MVP is a video benchmark, the paper does not include much visualization of it. The only visualization is in the figure on page 1, where three video pairs are provided and only one frame is presented for each video. I would suggest that the paper visualizes at least one sample for each of the four categories.

2. All the evaluation results are quantitative but not qualitative. It would be better if the paper could include qualitative results to show a) How the models leverage the shortcuts in existing benchmarks and b) How the models fail in MVP.

3. For four out of the nine video sources, the MVP benchmark directly uses the video pairs provided by the sources. However, the details about this part of videos are unclear, especially whether they share the same property as the newly matched video pairs. For example, do their video pairs share the same question while having opposite answers? How visually similar are these video pairs?

4. While the paper evaluates four types of shortcuts of existing video QA benchmarks, the proposed MVP benchmark is only designed to mitigate two of these shortcuts (language shortcut and single-frame shortcut). While it is acceptable to only address part of the shortcuts, the readers would be curious about whether the MVP benchmark also suffers from the video shortcut and the course-grained caption shortcut. However, the paper does not evaluate the video-only baseline or the Socratic LLM baseline.

**Audience:**

Yes

**Audience Explanation:**

This paper is about multimodal LLMs and video understanding. Both have a large population of audience.

**Broader Impact Concerns:**

No broader impact concerns.

**Claims And Evidence:**

Yes

**Claims Explanation:**

1. The paper claims that there are shortcuts in existing video QA benchmarks. This is supported by the evaluation results of a few baselines leveraging shortcuts.
2. The paper claims that the proposed MVP mitigates the shortcuts. This is supported by the detailed dataset curation process and the evaluation of language-only and single-frame baselines.
3. The paper claims that the state-of-the-art multimodal LLMs still struggle with seemingly simple physical reasoning tasks. This is supported by the evaluation on the MVP benchmark.

**Requested Changes:**

1. The performance of InternVL2.5-8B on MVP is 39.9% in Table 4 but 39.0% in Figure 1. Please fix the typo. Moreover, the abstract and introduction report the best performance of 40.2% on the entire MVP benchmark, but Figure 1 only reports the result on the mini version. This would make the readers confused. I would suggest reporting the performance on both versions in Figure 1.

2. In "VideoLLMs performance on dataset sub-tasks", Section 4, the paper only mentions a few numbers for a few models and sub-tasks. The paper should includes a complete table listing all the sub-task performance in either the main paper or the appendix.

3. Evaluate the video-only baseline or the socratic LLM baseline on MVP.

4. Include more figures to demonstrate the MVP benchmark and the qualitative results of the models.

---

> ### Author Response · Authors · 2025-09-22
> **Addressing main suggestions such as running more shortcut baselines on MVP**
>
> Dear Reviewer,
>
> We thank you for recognizing the value of our paper to the community, i.e. exposing and addressing shortcuts with extensive evaluation and a large-scale curation process. Your suggestions were also very helpful and we incorporated them!
>
> We will address each comment, as well as our respective changes to the paper and experiments. To make it easy to spot what changed in the updated paper PDF, we temporarily put new addition between red brackets “[[...]]”.
> Note: if you look at the main results Table 4 you will find that we added additional shortcut baselines on MVP (after a great suggestion from you). For now these were run on MVP-mini only to speed up experiments but we will run on the full dataset for the final paper.
>
> ### **Clear presentation of results/numbers:**
> Thank you for this suggestion, we have now unified the mentions of MVP-mini vs MVP-full more.
>
> ### **Include fine-grained sub-task performance in the appendix:**
> As suggested, we now include several appendix tables that break down MVP into many fine-grained categories (Appendix Section E). In addition, we now also elaborate in more detail in Section 4.1 “VideoLLMs performance on dataset sub-tasks” on where models are failing using these taxonomies.
>
> ### **Evaluate more shortcut baselines on MVP:**
> This particular suggestion will make our paper stronger and more coherent, thank you! So we take the same shortcut baseline we used to analyze existing benchmarks (Table 1) and apply them to MVP (Table 4). Specifically we add the missing video-only and Socratic LLM baseline. As noted above, for now this shows MVP-mini numbers and we will run on MVP-full for the final paper (the trend should be the same as it is a representative subset). We discuss the findings in Section 4.1.
>
> ### **Include more figures to demonstrate MVP and qualitative results of the models:**
> We now include 18 examples (2 for each original source in MVP) in the appendix as well as some examples of intuitive physics in the main paper, where models are either failing or partially succeeding.
> We are happy to include even more examples in the main paper if more are helpful to the reader.
>
> Overall, we hope this addressed your main concerns and suggestions, and that our edits in the paper reflect this adequately. We are happy to discuss further in the following days if you have any follow up questions..

---

### Review · Reviewer_D1CL · 2025-08-27

**Summary Of Contributions:**

This paper introduces the Minimal Video Pairs (MVP) benchmark, which aims to address the problem of score inflation in existing Video QA benchmarks where models achieve score inflation by exploiting superficial visual or textual cues. MVP contains approximately 55,000 multiple-choice video QA examples, focusing on physical world understanding. To effectively resist shortcut solutions, each sample in MVP contains a minimal-change pair: a visually similar video with opposite answers. The model can only score if it correctly answers both questions in the minimal-change pair at the same time. The paper analyzes the shortcut solutions of the existing MVBench benchmark and, by introducing MVP, reveals the huge gap in physical world understanding of current VideoLLMs, where human performance reaches 92.9%, while the state-of-the-art models only reach 40.2%.

### Strengths

1. The authors conduct an in-depth analysis of potential shortcut solutions across all 11 datasets in the popular benchmark, using simple baselines such as language-only models, single-frame/image models, and Socratic LLMs.
2. The authors introduce MVP, a video QA benchmark for physical world understanding. It features minimally diverse videos and is an order of magnitude larger than similar benchmarks, with approximately 55,000 samples.
3. The authors benchmark closed-source and open-source state-of-the-art models, revealing significant gaps in physical world understanding.

**Audience:**

Yes

**Audience Explanation:**

This work proposes a shortcut-aware video question answering benchmark (MVP), which systematically exposes the "shortcut" bias of the existing Video-LLM evaluation using minimal pairs and single-frame filtering; this is of direct value to TMLR readers who are interested in multimodal evaluation, benchmark design, and video understanding modeling.

**Broader Impact Concerns:**

The main ethical risks lie in licensing compliance and potential privacy issues associated with the re-annotation and redistribution of multi-source videos (including egocentric ones), as well as the representativeness and bias introduced by sample distribution. It is recommended to supplement the license matrix and privacy handling instructions for each data source. Another risk is that single-frame filtering relying on an MLLM ensemble may encode model bias into the data and induce overfitting of the minimal-pair architecture. This requires increased bias auditing and usage specifications.

**Claims And Evidence:**

No

**Claims Explanation:**

1. The paper mentions that the majority of minimal-change pairs were mined through an automated process. While this allows the benchmark to achieve a large scale, it may also introduce noise. For example, entailment detection in the Language Table involves several handcrafted rules, and the robustness and generalization of these rules warrant further investigation. The definition of "fuzzy subset" mentioned in Appendix B.2 is also somewhat vague. Errors in these automated steps could result in some minimal pairs not being truly "minimally different" or "mutually exclusive," thus affecting the validity of the benchmark.
2. Section 4 of the paper discusses the impact of the single-letter output format and tests the performance of Gemini 1.5 after adding a reasoning prompt, showing a slight performance degradation. However, for such a benchmark, prompt engineering may have a far greater impact on VideoLLMs' performance than this. Exploring more diverse prompting strategies (e.g., Chain-of-Thought, few-shot prompting, etc.) and their impact on model performance on MVP will help more comprehensively assess the model's true capabilities.
3. The paper evaluated multiple state-of-the-art VideoLLMs, but most performed poorly on MVP. While this highlights the challenge, future research may require a deeper analysis of the specific modes of failure of these models, such as whether they suffer from systematic deficiencies in spatial reasoning, temporal understanding, or intuitive physics. Currently, simply providing overall and subcategory scores may not be sufficient for understanding model weaknesses and guiding improvements.

**Requested Changes:**

1. In addition to the Minimal Pair Score, it is recommended to supplement single-question accuracy, pair agreement rate, uncertainty measures (such as temperature sampling variance), confidence-accuracy calibration curves, and report the pair difficulty distribution.
2. Randomly sample N pairs from each major category for double-blind verification, reporting the noise ratio and major ambiguity types (occlusion, language ambiguity, and video quality).
3. Use a non-LLM single-frame classifier (such as a classic action classifier or temporal consistency classifier) to compare with the current LLM ensemble, reporting the agreement rate and difference examples.
4. If affordable, expand the GPT-4o and Gemini evaluation to a representative subset of the MVP-full dataset, standardizing the frame rate, resolution, and prompt. Alternatively, provide performance curves for the inference budget to avoid biased conclusions due to "different parameters."
5. In Appendix B.2, provide more specific examples or descriptions of the "several hand-crafted rules" used in constructing the Language Table dataset. Also, provide a clearer explanation of the definition of the fuzzy_subset function used in pairing the CLEVRER dataset (particularly the meaning of "one mismatch"). It is recommended to supplement the results of a small-scale manual validation, such as randomly sampling minimal pairs generated by the automated process, and report the quality ratio of human judgment to quantify the reliability of the automated process.
6. Currently, the performance of models in the Intuitive Physics category is generally close to random. Please further analyze the specific reasons for this extremely low performance, for example, whether the models generally have systematic deficiencies in specific physical concepts such as object permanence, gravity, and collision dynamics. If possible, provide a qualitative analysis of the errors of the best-performing models in this category, and provide one or two specific error cases to help the community understand the specific patterns of model failure.
7. The exploration of prompt engineering in the paper is limited to the effects of Gemini 1.5 after the addition of reasoning prompts. Please also conduct similar reasoning prompt experiments for at least one open-source SOTA VideoLLM. Alternatively, briefly discuss other common prompting strategies (e.g., few-shot examples).

---

> ### Author Response · Authors · 2025-09-22
> **Addressing main suggestions such as providing additional metrics on top of Minimal Pair Score**
>
> Dear Reviewer,
>
> Thank you for your valuable and very detailed feedback. We will address each comment, and refer to respective changes to the paper or experiments. To make it easy to spot changes in the updated paper PDF, we highlight changes between red brackets “[[...]]”.
>
> ### **Additional metrics:**
> The reviewer suggested five additional metrics to our primary metric (Minimal Pair Score):
>
> 1. **Single-answer-accuracy & pair agreement**: We add two additional tables (Table 7 and 8) where we report both single-answer accuracy and pair agreement, and add a discussion in the paper.
> 2. **Uncertainty measures**: Most VideoLLMs have “do_sample=False” during text generation, so temperature is effectively 0; this makes sense in a multi-choice QA format (model only needs to produce a single letter response). However, we conduct an ablation with various prompting strategies and report these results in the paper; we observe little variance in downstream performance.
> 3. **Confidence-accuracy calibration curves:** We show the relation between confidence (logprob of the answer letter A vs B) and accuracy in Figure 6 of the updated paper.
> 4. **Pair difficulty distribution:** While MVP does not explicitly have a split such as “hard” vs “easy”, we add an analysis in the paper (p. 13) with three proxies for difficulty: video length on the vision side and word frequency/sentence difficulty on the text side.
>
> ### **Quantity and proportion of noisy or ambiguous dataset cases:**
> To quantify the level of noise in our data, we had already collected human accuracy on MVP (see Table 4). Additionally we now provide the explicit human accuracy for each of the 9 data sources in the appendix, and further analyze certain subsets in detail with manual inspection. This is added at the end of Section 3. Similarly, we expanded on the procedure for mining and constructing minimal pairs (e.g. Language Table) in the updated Appendix.
>
> ### **Replacing multimodal-LLM-ensemble for single-frame-filtering with a more traditional method:**
> While it is possible to explore alternatives for identifying single-frame-solvable examples, MVP involves open-world concepts, and therefore such action classifiers would similarly need to perform open-world concept discrimination. One option could be to leverage a Video-CLIP model non-parametrically, but we would expect performance to be worse and more sensitive to wording.
>
> ### **Expand the GPT-4o and Gemini evaluation to a representative subset, standardizing the frame rate, resolution, and prompt:**
> The GPT-4o and Gemini evaluations are conducted on MVP-mini, which is a representative subset of the full dataset, we will clarify this in the paper; we standardize the prompts across models.
> Regarding frame rate and resolution, API models are often trained with different default parameters and often provide (or internally apply) their own custom preprocessing to the video. We adopt the default parameters for each (API) model, but have now also conducted a small test on Gemini: we run a subset of examples from each category split of MVP with different frame numbers (8,16,32) as well as 1fps (6 different combinations), and find no difference in prediction. Only when we change Gemini’s default resolution “low”, do we see a significant drop in performance from 37.5% to 12.5%.
>
> ### **More detailed analysis of where model fails:**
> Specifically for the intuitive physics subsets you mentioned as hard for models, we provide a short discussion of recent literature in the updated paper (p. 9).
> We also expand our existing “Fine-grained failure analysis” section in Section 4.1 where we discuss various subsets, and show intuitive physics examples.
> Finally, we also note that we added additional shortcut baselines to the main results Table 4 on MVP.
>
> ### **More extensive prompting/CoT analysis:**
> With Gemini we observed that performance did not improve when allowing more time to reason before the answer.
> So we tested if the same trend would hold for our strongest open-source VideoLLM (InternVL2.5-8B) for two prompts:
> 1. The same as provided to Gemini, so allowing short reasoning of 1-3 sentences.
> 2.  A CoT prompt to reason for longer “step-by-step” and listing all relevant objects, key events etc.
>
> We find that CoT prompts only marginally helps, similar to Gemini (summarized in updated paper, Table 6).
> While few-shot prompting is very interesting, we did not explore it since the majority of models are never trained with more than 1 video as input, and requires extensive compute resources..
>
>
> Overall, we hope this addressed your main suggestions, and that our updated paper reflects this adequately. We are happy to discuss further in the following days and provide more evidence.

---

> > ### Author Response · Authors · 2025-09-28
> > **Discussion**
> >
> > Just a quick note to kindly encourage you to take a look at our rebuttal before the discussion period ends. We’ve carefully addressed all major points (both empirically or conceptually) and would appreciate your feedback on our clarifications and new results.

---

### Review · Reviewer_wWdq · 2025-09-03

**Summary Of Contributions:**

**Summary of Contributions:**
+ This work proposes the Minimal Video Pairs (MVP) benchmark, a large-scale shortcut-aware VideoQA benchmark for measuring the physical understanding and reasoning capabilities of video language models. It further introduces an automated pipeline to mine minimally different video pairs from existing video sources by leveraging visual embeddings and video meta-data.

+ Presents an analysis of potential shortcut solutions on the 20 tasks encompassed by MVBench benchmark using a variety of simple baselines.

+ Evaluate both open-source and closed-source SOTA models on MVP, signalling a gap in physical world understanding to human performance vs GPT4-o and Gemini.

**Additional Comments:**

Minor comments:
The caption for Figure 1 is not informative.

The presentation of the paper can be improved by including more intuitive examples and figures with self-explanatory captions.

**Audience:**

Yes

**Audience Explanation:**

The work is interesting and relevant to the large computer vision community, particularly those focused on video understanding.

**Broader Impact Concerns:**

The MVP benchmark raises broader impact concerns by potentially oversimplifying temporal reasoning through limited task formats and unclear shortcut definitions. Without addressing dataset noise and failure analysis, there is a risk of overestimating model competence, which could misguide the community and hinder progress toward robust, real-world video reasoning systems.

**Claims And Evidence:**

No

**Claims Explanation:**

+ While the paper excessively uses the term “shortcuts” and has also done a shortcut analysis on MVBench – it fails to coherently explain what a shortcut actually is. I understood this in much later in the paper. A well-motivated example with a figure could have made things more clear.

+ While the scale of MVP is huge as compared to prior attempts (VINOGROUND and TempCompass) it still restricts to multi-choice QA instead of also including other task formats thereby hindering the understanding of how temporal perception may vary across different tasks – a limitation pointed by these prior works themselves. (Cheng et.al.,V-STaR: Benchmarking Video-LLMs on Video Spatio-Temporal Reasoning, arXiv, 2025) tells that (what/when/where) fails and prompts the models to give localized instances of the same instead of simple binary answers. (Zhou et.al., TUMTraffic-VideoQA: A Benchmark for Unified Spatio-Temporal Video Understanding in Traffic Scenes, arXiv, 2025) shows that robust spatio-temporal reasoning requires multiple tasks like referred object captioning, etc. MVP’s minimal pair evaluation, however,  does not show why the model failed.

+ The paper misses mentioning the quantity of cases where the dataset examples are noisy / ambiguous. While the paper reports the accuracy, there is no failure case analysis as to why models fail on certain examples. A section on the same can make the contribution of MVP more intuitive.

+ The paper could have also included multiple examples of the minimal change pairs for the sake of seeing diversity across visual changes included in the video pairs

**Requested Changes:**

+ Provide a clear and coherent definition of “shortcuts”, supported by a well-motivated example and an illustrative figure.

+ Expand the benchmark beyond multi-choice QA to include additional task formats (e.g., captioning, localization, temporal ordering) to better capture temporal perception across diverse tasks. (Optional)

+ Discuss related findings from V-STaR (Cheng et al., 2025) and TUMTraffic-VideoQA (Zhou et al., 2025) to highlight the need for richer task types and failure localization, and clarify how MVP addresses or differs from these limitations.

+ Explicitly mention the quantity and proportion of noisy or ambiguous dataset cases, along with steps (if any) taken to mitigate their effect.

+ Add a failure case analysis section explaining why models fail on certain examples, complementing the reported accuracy metrics.

+ Include multiple diverse examples of minimal change pairs in the main text or appendix to showcase the range of visual variations represented in the dataset.

---

> ### Author Response · Authors · 2025-09-22
> **Addressing main suggestions such as more detailed failure analysis**
>
> Dear Reviewer,
>
> Thank you for your valuable and very nuanced feedback with pointers to recent relevant papers!
>
> We will address each comment, as well as our respective changes to the paper and experiments. To make it easy to spot changes in the updated paper PDF, we highlight changes between red brackets “[[...]]”.
>
> ### **Define & illustrate what “shortcut” means:**
> We added a brief subsection “2.1 Shortcut solutions in video understanding” to the paper where we define shortcuts and common ways to detect them in our paper’s context, as well as provide a short illustrative figure:
>
> *What is classified as a shortcut depends on the skills a task is meant to test, e.g., whether a feature is \textit{causal} or merely \textit{spurious} \citep{geirhos2020shortcut}; detecting and mitigating shortcut learning therefore remains an open problem \citep{geirhos2020shortcut}.
> In our setting, …*
>
> ### **Additional task formats (captioning, localization, temporal ordering) and analyzing failures:**
>
> Other task formats, such as localization, have distinct advantages that may allow pinpointing the exact temporal or spatial failures (Cheng et.al.,V-STaR: Benchmarking Video-LLMs on Video Spatio-Temporal Reasoning, arXiv, 2025), however they do not allow the same ease of adoption and flexibility: applying a VideoLLM to such task (e.g. V-STaR) requires careful attention to the CoT prompts for different what/when/where sub-tasks, and involves a mix of more complex metrics such as “mean visual Intersection over Union.” Designing MVP as a multiple-choice QA task ensures a simple unified setup that is easy to adopt by the community. At the same time, multiple-choice QA is highly flexible since most tasks can be reformulated as QA; note that many videos we source examples from were not originally designed for QA, such as Language Table.
>
> Our work is complementary to V-STAR. We provide a framework for gathering minimal pairs and a concrete large-scale benchmark to address shortcuts. On the other hand,V-STaR does not leverage minimal change pairs, but instead decomposes questions into sub-steps e.g. what/when/where to build insight into models’ reasoning process. While V-STaR’s fine-grained grounding in spatial or temporal position helps locate where a model fails, it would not strictly control for spurious cues which is the focus of our work.  Finally, TUMTraffic-VideoQA is a valuable resource for advanced video understanding in complex/chaotic real-world scenarios. Orthogonal to this work, we primarily rely on sources that test more atomic “simpler” skills, that might perhaps not require chaining together as many tasks and reasoning steps but instead measure for a basic  understanding of intuitive physics and simple physical interactions.
>
> To make all this more clear to the reader, we now describe the advantages and disadvantages of task formats to better motivate our decision in the paper at the beginning of Section 3.1 (“Task Definition”).
>
> ### **Quantity and proportion of noisy or ambiguous dataset cases:**
>
> To quantify the level of noise in our data, we had collected human accuracy on MVP reported in the main results Table 4. Additionally we now provide the explicit human accuracy for each of the 9 data sources in the appendix, and further analyze certain subsets in detail: We conduct a small manual annotation on the examples humans struggled with to quantify non-mutually exclusive answer options (a rare issues stemming from our automated pipeline). This is added at the end of Section 3. To summarize, we find that in some rare cases (less than 10% of all examples) two answer candidates can both be true at the same time, or otherwise that the question is ambiguous.
>
> ### **More detailed analysis of where model fails:**
> Specifically for the intuitive physics subsets, we provide a short discussion of recent literature in the updated paper (p. 9). We also expand our existing “Fine-grained failure analysis” section in Section 4.1 where we discuss various low subsets that stand out, and show some examples from one of the intuitive physics subsets (GRASP).
> Finally, while not explicitly requested by this reviewer, we also note that we added additional shortcut baselines to the main results Table 4 on MVP, in line with our shortcut analysis of existing benchmarks: the video-only and Socratic LLM baselines, as well as a third single-frame baseline.
> (For now these were run on MVP-mini only to speed up experiments but we will run on the full dataset for the final paper.)
>
> ### **Illustrate diversity and phenomena in MVP benchmark with examples in appendix:**
>
> We now added 18 examples in the appendix, two samples for each of the nine sources we use. For STAR examples we covered faces for privacy reasons.
>
> We have addressed your main concerns and suggestions, and have updated the paper to reflect these changes. We are happy to discuss further and address any remaining questions in the following days.

---

> > ### Author Response · Authors · 2025-09-28
> > **Discussion**
> >
> > Just a quick note to kindly encourage you to take a look at our rebuttal before the discussion period ends. We’ve carefully addressed all major points (both empirically or conceptually) and would appreciate your feedback on our clarifications and new results.

---

### Decision · Action_Editor_rgCN · 2025-11-05

**Recommendation:** Accept with minor revision

**Additional Comments:**

The presentation of the paper still has room for improvement. For example, when introducing shortcuts, it would be helpful to include a concrete example with an illustration to aid the reader's understanding of what a shortcut is.

**Audience:**

Yes

**Audience Explanation:**

The study contributes to the growing body of research on shortcut-aware evaluation in multimodal reasoning, specifically targeting video-language models’ physical understanding. Given the current momentum in video reasoning benchmarks and model evaluation, the MVP dataset and findings on shortcut mitigation would be of clear interest to researchers in vision-language modeling, dataset curation, and evaluation methodology.

On the other hand, the downside is that there are tons of video QA benchmarks proposed nowadays and many of them also aim to mitigate the shortcuts. In this sense, this paper lacks a standout contribution among similar datasets.

**Claims And Evidence:**

Yes

**Claims Explanation:**

The paper provides reasonable experimental evidence through systematic evaluations on both open- and closed-source video-language models using the proposed Minimal Video Pairs (MVP) benchmark. The revised version includes improved visualization, noise analysis, and additional baselines (e.g., Video-only, Socratic LLM), which strengthen the empirical support for its claims. Although some limitations remain in scope and temporal grounding, the overall claims are clearly supported by data and analyses.

---

> ### Author Response · Authors · 2025-11-17
> **Camera Ready submitted**
>
> Dear AE,
>
> Thank you for these good news and the additional higher-level feedback! We just submitted the camera-ready version of our paper with the minor revision you suggested:
>
> *For example, when introducing shortcuts, it would be helpful to include a concrete example with an illustration to aid the reader's understanding of what a shortcut is.*
>
> We therefore added an example for each of the 4 shortcut baselines we study.
>
> On a final note, we agree that "there are tons of video QA benchmarks proposed nowadays and many of them also aim to mitigate the shortcuts". However, I personally see the value of this work in providing a unified + diverse + large-scale collection of minimal video pairs. Many dataset do not try to curate minimal pairs of video or only do so in a smaller scale and not across many existing datasets. Let's see what datasets will end up being useful to the community!